

# BOREAS - a new MAX-DOAS profile retrieval algorithm for aerosols and trace gases

Tim Bösch[1], Vladimir Rozanov[1], Andreas Richter[1], Enno Peters[1,a], Alexei Rozanov[1], Folkard Wittrock[1], Alexis Merlaud[4], Johannes Lampel[3,9], Stefan Schmitt[3], Marijn de Haij[2], Stijn Berkhout[6], Bas Henzing[5], Arnoud Apituley[2], Mirjam den Hoed[2], Jan Vonk[6], Martin Tiefengraber[7,8], Moritz Müller[7,8], and John Philip Burrows[1]

[1]Institute of Environmental Physics, University of Bremen, Bremen, Germany
[2]Royal Netherlands Meteorological Institute (KNMI), De Bilt, the Netherlands
[3]Institute of Environmental Physics, University of Heidelberg, Heidelberg, Germany
[4]Royal Belgian Institute for Space Aeronomy (BIRA-IASB), Brussels, Belgium
[5]Netherlands Organization for Applied Scientific Research (TNO), Utrecht, the Netherlands
[6]National Institute for Public Health and the Environment (RIVM), Bilthoven, the Netherlands
[7]LuftBlick, Kreith, Austria
[8]Institute of Atmospheric and Cryospheric Sciences, University of Innsbruck, Innsbruck, Austria
[9]Airyx GmbH, Eppelheim, Germany
[a]now at: Institute for protection of maritime infrastructures, Bremerhaven, Germany

*Correspondence to:* Tim Bösch (tim.boesch@iup.physik.uni-bremen.de)

**Abstract.** We present a new MAX-DOAS profiling algorithm for aerosols and trace gases, BOREAS, which utilizes an iterative solution method including Tikhonov regularization and the optimal estimation technique. Performance tests are separated into two parts. First, we address the general sensitivity of the retrieval on the example of synthetic data calculated with the radiative transfer model SCIATRAN. In a second part of the study, we demonstrate BOREAS profiling accuracy by validating results

with the help of ancillary measurements carried out during the CINDI-2 campaign in Cabauw, the Netherlands in 2016.

The synthetic sensitivity tests indicate, that the regularization between measurement and a priori constraints is insufficient when knowledge of the true state of the atmosphere is poor. We demonstrate a priori pre-scaling and extensive regularization tests as a tool for the optimization of retrieved profiles. The comparison of retrieval results with in situ, ceilometer, $NO_2$ LIDAR, sonde and long path DOAS measurements during the CINDI-2 campaign always shows high correlations with coefficients

greater than 0.79. The largest differences can be found in the morning hours, when the planetary boundary layer is not yet fully developed and the concentration of trace gases and aerosol, as a result of a low night-time boundary layer having formed, is focused in a shallow, surface near layer.

## 1 Introduction

Aerosols and trace gases play an important role for life on Earth as high concentrations have adverse impacts on human health

and climate. Furthermore, aerosols impact on Earth's energy budget by radiative forcing. They interact with solar radiation by scattering and absorption processes. Additionally, they have an impact on the formation of clouds when acting as cloud con-





densation nuclei (CCN). Despite considerable efforts by the scientific community, uncertainties on aerosol radiative forcing are still high (Stocker, 2014).

In addition to aerosols, atmospheric trace gases are of importance in an increasingly urbanized world as they impact on human health, agriculture, acid deposition, and climate (e.g. Lelieveld et al. (2002)). Trace gases in the troposphere such as nitrogen

dioxide and sulfur dioxide ($NO_2$, $SO_2$), nitrous acid (HONO), formaldehyde (HCHO) and glyoxal (CHOCHO) need to be measured to assess their impact on air quality and because their spatial and temporal inhomogeneous distribution provides information on different emission sources e.g. the combustion of fossil fuels, biomass burning and agriculture (Wittrock et al., 2004; Irie et al., 2011; Pinardi et al., 2013; Hendrick et al., 2014). The separation and identification of anthropogenic from natural sources as well as measurements of their temporal variability improves the understanding of their effects on climate

and life on Earth. The current lack of knowledge leads to uncertainties in temporal and spatial emission patterns of trace gases and aerosols and limits our ability to fully understand atmospheric processes. This results in a need for comprehensive measurements to fill these gaps.

For more than 15 years, Multi-Axis Differential Optical Absorption Spectroscopy (MAX-DOAS) measurements have been used to investigate the chemical composition of the troposphere (Hönninger and Platt, 2002; Bobrowski et al., 2003; Leser H.

et al., 2003; Wittrock et al., 2004; Wagner et al., 2004; van Roozendael et al., 2004). This passive remote sensing method is based on absorption spectroscopy applied to measurements of scattered sunlight. The advantages of ground-based MAX-DOAS when compared to satellite observations are the high sensitivity for the lowermost layers of the troposphere, the high temporal and spatial resolution of measurements, and the lower cost.

The strength of the absorption signal detected in scattered sunlight depends on the absorber amounts and their vertical distri-

bution but also on the length of the light path. In general, this length in a certain altitude layer is a function of the measurement geometry. Therefore, a set of MAX-DOAS radiance measurements taken at different elevation angles (lines of sight, LOS) contains information about the vertical distribution of trace gases, which can be retrieved. However, the retrieval of absorber profiles from MAX-DOAS measurements is an ill posed problem that needs additional constraints (see Section 3.2). Thus, the inversion, which is applied for retrieving vertical profile concentrations, is done by elaborated mathematical methods such as

optimal estimation (OE). In addition to the trace gas profile, the aerosol extinction profile needs to be retrieved as well as it has a non-linear effect on the trace gas retrieval and it is too variable to be approximated by climatologies. In contrast to the trace gas retrievals, aerosol retrievals are strongly non-linear, necessitating iterative inversion schemes such as the Gauss-Newton algorithm or the Levenberg Marquardt algorithm (Rodgers, 2004).

Profile retrieval algorithms used in the scientific community are either inversion algorithms or parametrized approaches. Inver-

sion algorithms directly link the measurement quantity to the vertical profile of the target absorber with the help of a forward model (Wittrock et al., 2004; Hendrick et al., 2004; Wagner et al., 2004; Frieß et al., 2006; Wittrock, 2006; Clémer et al., 2010; Wang et al., 2017). This model is usually calculated with a radiative transfer model (RTM) assuming a selected set of atmospheric conditions. For trace gases, the model also includes a sensitivity matrix which can be considered as derivative of measurement quantity with respect to the trace gas concentration for each altitude level. For aerosols, this sensitivity matrix is

normally calculated via perturbation theory. This perturbation approach is shortly summarized as follows: by consideration of



a trace gas with a well known vertical distribution such as the oxygen dimer $O_2$-$O_2$ (or short $O_4$), the aerosol extinction for each layer is changed gradually and the resulting modelled observations for different LOS are compared with measurements until the iteration converges (e.g. the difference between measurement and modelled quantity is small enough).

Parametrization on the other hand means, that a forward model, based on a limited set of parameters, is used to describe the
measurement quantity. Frequently used forward model parameters are integrated values and profile information such as shape and height of certain absorber layers (Sinreich et al., 2005; Lee et al., 2009; Li et al., 2010; Vlemmix et al., 2011; Wagner et al., 2011; Sinreich et al., 2013). The forward model results are calculated with an RTM and are least squares fitted to the measurements. Generally, look up tables (LUT) are pre-calculated for a set of different scenarios, parameters and geometries covering all relevant atmospheric conditions, avoiding high computational efforts during near real time calculations.

Inversion algorithms have the advantage that they are not limited to the scenarios used when creating the LUT but unrealistic profiles are possible when the measurement, inversion, or regularization (weighting between the information from measurements and a priori informations) is poor. On the other hand, parametrized approaches evaluate profiles much faster, as the slow forward model computation was already done when creating the LUT.

Although efforts for deriving concentration profiles from MAX-DOAS measurements have been made for more than one de-
cade, profiling is still a difficult task and results of different algorithms can differ strongly when the absorber of interest is highly variable and inhomogeneous on spatial and temporal scales (Zieger et al., 2011; Vlemmix et al., 2015; Frieß et al., 2016). Comparison studies of different profiling algorithms for synthetic as well as for real data summarizing the current state of the art and including results from the algorithm introduced here will be reported by Frieß et al. (2018) and Tirpitz et al. (2018).

The purpose of this study is the introduction of IUP Bremen's new MAX-DOAS profile retrieval algorithm for aerosols and trace gases BOREAS (Bremen Optimal estimation REtrieval for Aerosols and trace gaseS) which has been developed to improve on the earlier profile retrieval algorithm (Wittrock, 2006). BOREAS uses a novel approach for the retrieval of aerosols but a similar optimal estimation technique for the retrieval of trace gases. In contrast to perturbation based inversion algorithms, BOREAS uses the change (from an a priori state) of depth in an absorption band to get information on the aerosol content which
caused this change. To the authors' knowledge, this approach has never been used within an operational profiling algorithm and it complements the variety of methods with another promising technique. The development of BOREAS aimed at several key properties:

1. Flexibility: the algorithm should retrieve aerosol and trace gas profiles from any MAX-DOAS measurement with pre-filtering options for the data.

2. Accuracy/stability: the algorithm should be stable in terms of varying atmospheric conditions when retrieving profiles for several years of data. The profiling results (modelled observation) should fit the measured observations with a high accuracy.

3. Automation: the retrieval should respond to problematic data/settings (e.g. low information content, wrong regularization) automatically with an included problem solution scheme.





4. Fast: the algorithm should be fast enough to allow near real time profile retrievals.

The manuscript is organised as follow: in the first section, a typical MAX-DOAS measurement and its information content are shortly described. The next section focuses on the theoretical background of the retrieval algorithm. This section is followed by a detailed sensitivity study. The 4th section is divided into an analysis of synthetic data, a discussion of error sources and an

example of application to real measurements which shows that BOREAS is able to retrieve profiles with high reliability and accuracy. The final section 5 concludes and summarizes the results presented in this study.

## 2   MAX-DOAS measurements and the DOAS retrieval

Modern ground-based MAX-DOAS instruments are capable of measuring scattered sunlight in the ultraviolet (UV) and visible (VIS) spectral range with a full azimuthal and elevation angle coverage of the hemisphere (see Figure 1). In general,

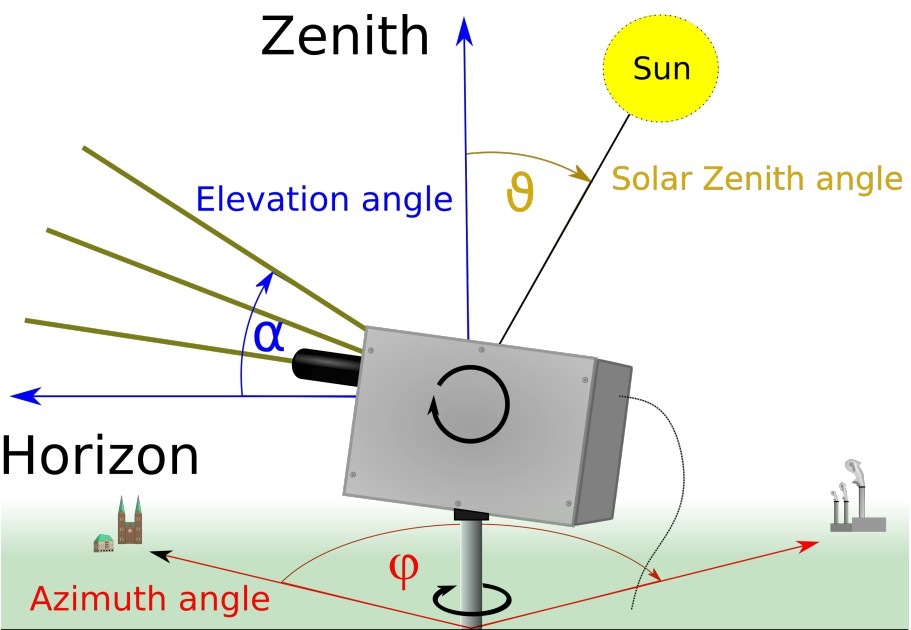

**Figure 1.** Schematic representation of typical MAX-DOAS measurement geometries.

the spectral least-squares fit of absorption cross sections and a polynomial to the slant optical thickness (logarithm of a sun normalized radiance at a certain LOS, also called slant optical thickness, SOT) $\tau = -\ln(I/I_0)$ (DOAS fit) provides integrated concentrations $\rho$ along the light path L, the so called slant column densities S (or SCD) (Platt and Stutz, 2008).

$$S = \int_0^L \rho(s)ds \qquad (1)$$

The length of the light path L depends on the sun position, the scattering properties of the atmosphere and on the viewing

geometry. For ground based measurements, the extraterrestrial solar spectrum $I_0$ is usually replaced by an intensity spectrum





measured in zenith direction $I_{ref}$ (reference spectrum) to compensate for Fraunhofer lines and instrumental issues (e.g. temperature dependent wavelength shifts). As this zenith spectrum itself is usually contaminated by the absorber of interest to some extent, the difference of the two SCDs, $\Delta S$ is retrieved and referred to as differential SCD. The differential optical thickness $\Delta \tau_j$ of a certain absorber j only, is defined as

$$\Delta \tau_j(\lambda, \mathbf{\Omega}) = \sigma_j(\lambda) \Delta S_j(\mathbf{\Omega}), \tag{2}$$

where $\sigma_j(\lambda)$ is the spectral absorption cross section and $\mathbf{\Omega} = \{\alpha, \varphi, \vartheta\}$ represents the angular variables. Here, $\alpha$ is the elevation angle (or LOS), $\varphi$ is the relative azimuth angle (RAA) between sun and viewing azimuth, and $\vartheta$ is the solar zenith angle (SZA). When using a single zenith spectrum measured around noon as $I_{ref}$, the tropospheric signal in $I_{ref}$ is minimised but for absorbers which are also present in the stratosphere, stratospheric absorption contributes significantly to $\Delta S$ in the morning and evening. This is due to diurnal variations in the stratospheric amounts of some absorbers such as $NO_2$ and the change in light paths through the stratosphere because of a varying SZA (see e.g. Leser H. et al. (2003)). To minimise the stratospheric signal, a reference spectrum measured close in time to the intensity I for a certain geometry is applied. Usually, a scan of measurements in different elevation angles is followed by a measurement in zenith direction so that the whole scan can be fitted with the same reference spectrum.

The ratio of slant column density S and the vertically integrated absorber concentration $V = \int_0^H \rho(z) dz$ (vertical column density V) is called air mass factor M (AMF).

$$M(\mathbf{\Omega}) \equiv \frac{S(\mathbf{\Omega})}{V} \tag{3}$$

Generally, this value depends on the radiative transfer through the atmosphere and on parameters such as SZA, surface albedo, wavelength, and the profiles of aerosols and gaseous absorbers. For trace gases in the boundary layer, it can in first approximation be described by the geometric air mass factor

$$M_{geom}(\alpha) = \frac{S(\alpha)}{V} = \frac{1}{\sin(\alpha)}, \tag{4}$$

assuming that the trace gas layer is located below the last scattering point (Hönninger et al., 2004). The differential AMF (dAMF), assuming a zenith reference spectrum, is then $\Delta M = S/V - S_{ref}/V = (1 - \sin(\alpha))/\sin(\alpha)$. The difference between geometric and true AMF depends strongly on the wavelength, elevation angle, relative azimuth angle and aerosols.

Because of the altitude depending concentration of absorbers, there is a need for a similar quantity describing the ratio of $S$ and $V$ in a certain layer. The concept of box air mass factor $\mathcal{M}_z$ (BAMF) introduces this height dependence by defining the ratio for each altitude layer z:

$$\mathcal{M}_z(\mathbf{\Omega}) = \frac{S_z(\mathbf{\Omega})}{V_z} \tag{5}$$

Here, $\mathcal{M}_z$ depends on the altitude in contrast to the total AMF $M$. Box air mass factors are calculated by radiative transfer models and can be understood as sensitivity of the partial slant column to the partial vertical column at a certain altitude. On the other hand, multiplication of a profile of partial vertical columns with $\mathcal{M}_z$ leads to the corresponding slant column density.



# 3 Retrieval algorithm description

The profile retrieval algorithm BOREAS was developed in order to retrieve aerosol and trace gas vertical profiles from MAX-DOAS measurements. The aerosol retrieval is fully implemented within the RTM SCIATRAN (Rozanov et al., 2014) to decrease computation time for the iterative minimization scheme. BOREAS is a Python written analysis script which calls

5   SCIATRAN for the aerosol retrieval and for calculations of BAMF matrices which are then used within an optimal estimation based trace gas retrieval. A flow chart depicting BOREAS is shown in Figure 2. In the next subsections, we give an overview of the individual steps of the algorithm.

**Figure 2.** BOREAS analysis flow chart. The algorithm is separated into two steps. Step 1: Aerosol retrieval within a Tikhonov-Regularization. Step 2: Optimal estimation based trace gas retrieval.





### 3.1 Retrieval of aerosol profiles

The standard DOAS fit does not provide direct information about aerosols present in the atmosphere. However, the scattering and absorption properties of aerosols have an impact on the measured differential slant column density $\Delta S(\mathbf{\Omega})$ because scattering processes can significantly modify the light path. In general, the absorption of photons by aerosols plays a small role in

contrast to scattering effects, which lead to the modification in photon path length. The single scattering albedo $\omega$ (SSA)

$$\omega = \frac{\sigma_s}{\sigma_e} = \frac{\sigma_s}{\sigma_s + \sigma_a} \tag{6}$$

quantifies the ratio of scattering $\sigma_s$ to total extinction efficiency $\sigma_e$. Here, $\sigma_e$ is the sum of scattering and absorption efficiency $\sigma_e = \sigma_s + \sigma_a$. For urban pollution, SSA values are in the range of 0.90 to 0.99 for the visible spectral range (see e.g. Dubovik et al. (2002)) and they stay more or less constant when the aerosol type does not change over time and altitude. Therefore,

the SSA is not a quantity to be retrieved in BOREAS and is kept constant with typical values for the prevalent aerosol type (e.g. urban pollution: $SSA = 0.92$). The angular scattering distribution of aerosols is fully quantified by the scattering phase function. Within BOREAS, we use the Henyey-Greenstein approximation (Henyey and Greenstein, 1941) with constant values for the asymmetry factor g which quantifies the amount of forward and backward scattering (e.g. urban pollution: $g = 0.68$). In this parametrization, the optical properties of aerosols are fully defined with g, SSA and an extinction coefficient profile $\sigma_e(z)$.

The latter is the retrieval parameter for BOREAS aerosol retrievals. Note, that the usage of measured phase functions will be implemented in BOREAS in the future, since Henyey-Greenstein is in some situations not an accurate representation of the atmospheric aerosol scattering distribution.

If $\sigma_e(z)$ is known, then the aerosol optical thickness (AOT) can be determined as $\tau_{aer} = \int_0^H \sigma_e(z)dz$, where the integration is performed over the entire atmosphere. If the vertical profile of an absorber number density is known, differences between

modelled and measured S for this absorber are the result of differences between the assumed and real aerosol profile. As observations at different LOS have varying sensitivity to the presence of aerosols at different altitudes, this can be used to retrieve an aerosol profile.

Generally, the vertical profiles of species in the troposphere are unknown because of the temporal and spatial variability of emission sources, transport and conversion processes. However, the oxygen monomer $O_2$ is in this respect an exceptional

species because it only depends on pressure and temperature. Furthermore, as the oxygen dimer $O_4$ concentration is proportional to the squared monomer concentration, its profile is exponentially decreasing with altitude as well. The $O_4$ slant column density can easily be determined, because $O_4$ has spectral absorption features in the wavelength regions of most DOAS fits. Detailed sensitivity studies of $O_4$ $\Delta S(\mathbf{\Omega})$ measurements regarding changes in atmospheric aerosol properties can be found in Wagner et al. (2004) and Frieß et al. (2006).

In the BOREAS aerosol retrieval algorithm, the difference between modelled and measured $O_4$ differential slant optical thickness's $\Delta\tau(\lambda, \mathbf{\Omega})$ (DSOT) is used to retrieve aerosol extinction profiles in an iterative process. The measured $\Delta\tau(\lambda, \mathbf{\Omega})$ is calculated using Eq. (2), where $\Delta S_i(\mathbf{\Omega})$ is the DSCD retrieved in the framework of the standard DOAS fit, and the simulated $O_4$ DSOT ($\Delta\tilde{\tau}(\lambda, \mathbf{\Omega})$) is calculated as follows:





$$\Delta\tilde{\tau}(\lambda,\boldsymbol{\Omega},N_a(z)) \ = \ \ln\frac{I(\lambda,\boldsymbol{\Omega},N_a(z))}{I_{ref}(\lambda,N_a(z))}\,, \tag{7}$$

where $I(\lambda,\boldsymbol{\Omega},N_a(z))$ and $I_{ref}(\lambda,N_a(z))$ are the intensities calculated using SCIATRAN under the assumption that $O_4$ is the only absorber. The dependency on the reference geometry $\boldsymbol{\Omega}_{ref}$ is summarized with the index $ref$ and will be neglected from

now on. The vertical profiles of pressure and temperature are used according to the US standard atmosphere model (NASA, 1976) but can be replaced by measured atmospheric conditions when available. The aerosol loading is described using an a priori concentration number density vertical profile $N_a(z)$.

The inverse problem with respect to the aerosol optical depth is then formulated as

$$\left\|\Delta\tau(\lambda,\boldsymbol{\Omega}) \ - \ \Delta\tilde{\tau}(\lambda,\boldsymbol{\Omega},N_a(z)) \ - \ P(\lambda,\boldsymbol{\Omega})\right\|^2 \ \longrightarrow \ \min\,, \tag{8}$$

where $P(\lambda,\boldsymbol{\Omega})$ is a polynomial of lower order and its argument $\boldsymbol{\Omega}$ emphasises that polynomial coefficients depend on the LOS angle. The assumptions made in this formulation are that 1) the $O_4$ absorption derived from the measurements does not depend on the concentration of other trace gases and 2) that the optical depth in an atmosphere without other absorbers can be described as the sum of the $O_4$ optical depth and a low order polynomial.

Since $\Delta\tilde{\tau}(\lambda,\boldsymbol{\Omega},N_a(z))$ is a non-linear and complicated functional of $N_a(z)$, an analytical solution of this minimization pro-

blem does not exist. To simplify the solution of the minimization problem given by Eq. (8), let us consider the variation of DSOT caused by the variation of the aerosol number density. The variation of DSOT in a linear approximation can be represented in the form of the following functional Tailor series (Rozanov and Rozanov, 2010):

$$\Delta\tilde{\tau}(\lambda,\boldsymbol{\Omega},N_a(z)) \ = \ \Delta\tilde{\tau}(\lambda,\boldsymbol{\Omega},\bar{N}_a(z)) \ + \ \int\limits_0^H W(\lambda,\boldsymbol{\Omega},z)\,\delta N_a(z)\,dz \ + \ \varepsilon_{lin}(\lambda)\,, \tag{9}$$

where

$$W(\lambda,\boldsymbol{\Omega},z) \ = \ \left.\frac{\delta\,\Delta\tilde{\tau}(\lambda,\boldsymbol{\Omega},N_a(z))}{\delta\,N_a(z)}\right|_{\bar{N}_a}$$

is a functional derivative of DSOT with respect to the aerosol number density profile $N_a(z)$ around the initial guess $\bar{N}_a(z)$. $W(\lambda,\boldsymbol{\Omega},z)$ is usually referred to as weighting function and $\varepsilon_{lin}(\lambda)$ is the linearisation error.

It follows that the weighting function provides a linear relationship between the variation of DSOT and the variation of an aerosol number density $N_a$ around the initial guess $\bar{N}_a$. Substituting now Eq. (9) into (8) and approximating the integral with

a finite sum, we have

$$\left\|\Delta\tau(\lambda,\boldsymbol{\Omega}) \ - \ \Delta\tilde{\tau}(\lambda,\boldsymbol{\Omega},\bar{N}_a(z)) \ - \ \sum_{k=1}^L \mathcal{W}(\lambda,\boldsymbol{\Omega},z_k)\,(x_k-\bar{x}_k) \ - \ P(\lambda,\boldsymbol{\Omega})\right\|^2 \longrightarrow \ \min\,. \tag{10}$$

Here, $\Delta x = x_k-\bar{x}_k = N_a(z_k)-\bar{N}_a(z_k)$, $L$ is the number of altitude levels, and $\mathcal{W}(\lambda,\boldsymbol{\Omega},z_k)$ is defined to satisfy the trapezoidal integration rule.




The solution of the minimization problem given by Eq. (10) is performed on a discrete wavelength grid. In fact, only five wavelengths over the $O_4$ absorption band centred at e.g. 477 nm are used.

The minimization problem can be reformulated in the following vector-matrix form:

$$\left\| \Delta\mathbf{y} - \mathbf{K}\Delta\mathbf{x} \right\|^2 \longrightarrow \min . \tag{11}$$

Here, the vector $\Delta\mathbf{y} = \mathbf{y} - \bar{\mathbf{y}} = |\Delta\mathbf{y}_1, \Delta\mathbf{y}_2, ..., \Delta\mathbf{y}_N|^T$ has the dimension $N \cdot M \times 1$ and describes the difference between measured and simulated DSOT at $N$ LOS angles and $M$ wavelengths. The $m$-th element of vectors $\mathbf{y}_n$ and $\bar{\mathbf{y}}_n$ is the differential slant optical depth of $O_4$ at wavelength m, given by

$$\{\mathbf{y}_n\}_m = \Delta\tau^-(\lambda_m, \boldsymbol{\Omega}_n), \ m = 1,...,M , \tag{12}$$

$$\{\bar{\mathbf{y}}_n\}_m = \Delta\tilde{\tau}^-(\lambda_m, \boldsymbol{\Omega}_n, \bar{N}_a(z)) , \tag{13}$$

where the superscript "-" denotes that a polynomial is subtracted. The state vector $\Delta\mathbf{x}$ has the dimension $L \times 1$ and matrix $\mathbf{K}$ has the dimension $N \cdot M \times L$, i.e., the matrix consists of $N \cdot M$ rows and $L$ columns. The $r$-th row of matrix $\mathbf{K}$ is then given by

$$\{\mathbf{K}\}_r = \left\{ \mathcal{W}^-(\lambda_m, \boldsymbol{\Omega}_n, z_1), \mathcal{W}^-(\lambda_m, \boldsymbol{\Omega}_n, z_2), ..., \mathcal{W}^-(\lambda_m, \boldsymbol{\Omega}_n, z_L) \right\} \quad \text{with} \quad r = (n-1)M + m \tag{14}$$

and contains weighting functions for the $m$-th wavelength, $n$-th LOS in $L$ layers.

The minimization problem given by Eq. (11) is solved employing an iterative Tikhonov regularization technique (Rodgers, 2004):

$$\mathbf{x}_{i+1} = \mathbf{x}_0 + (\mathbf{K}_i^T \mathbf{S}_y^{-1} \mathbf{K}_i + \mathbf{S}_0^{-1} + \gamma \mathbf{S}_t^T \mathbf{S}_t)^{-1} \mathbf{K}_i^T \mathbf{S}_y^{-1} (\mathbf{y} - \mathbf{y}_i + \mathbf{K}_i(\mathbf{x}_i - \mathbf{x}_0)) \tag{15}$$

Here, $i$ is the iteration number, $\mathbf{S}_0$ and $\mathbf{S}_y$ are the a priori and measurement covariance matrices, $\mathbf{K}_i$ is the weighting function matrix calculated using the estimated aerosol number density profile $\mathbf{x}_i$, $\mathbf{S}_t$ is the first order derivative matrix with Thikonov parameter $\gamma$ and $\mathbf{x}_0 = \bar{\mathbf{x}}$ and $\mathbf{y}_0 = \bar{\mathbf{y}}$ are the a priori profile and the representing a priori measurement vector, respectively.

There are three criteria to stop the iteration process:

- Convergence in parameter space, i.e., the maximum difference between the components of the state vector at two subsequent iterative steps does not exceed the selected criterion (e.g. $0.0001\,\text{km}^{-1}$).

- The root mean square difference between measured and simulated $O_4$ DSOT is less than selected (e.g. 0.001).

- The maximum number of iterations is reached.

In addition to the equations above, we introduce here other quantities which are useful to describe the retrieval. The gain matrix

$$\mathbf{G}_i = (\mathbf{K}_i^T \mathbf{S}_y^{-1} \mathbf{K}_i + \mathbf{S}_0^{-1} + \gamma \mathbf{S}_t^T \mathbf{S}_t)^{-1} \mathbf{K}_i^T \mathbf{S}_y^{-1} \tag{16}$$

describes the sensitivity of the solution to the measurement. Note, that this formulation includes also the Tikhonov term, leading to differences for $\gamma \neq 0$ with the definition given in Rodgers (2004). The averaging kernel $\mathbf{A}_i = \mathbf{G}_i \mathbf{K}_i$ (AK) characterizes the sensitivity of the solution to the true state. The trace of AK quantifies the degrees of freedom of the signal $d_s = \text{tr}(\mathbf{A})$ (DOF). This quantity is commonly understood as the number of individual pieces of information which, can be retrieved.



## 3.2 Retrieval of trace gas concentration profiles

Compared to aerosols, the inverse problem for trace gases is easier to solve because, under the assumption of an optically thin atmosphere, the relationship between trace gas concentration and measured differential slant column density is linear. Then, the forward model $\mathbf{F}(\mathbf{x}, \mathbf{b})$ is equal to a set of measurements $\mathbf{y}$,

$$\mathbf{y} = \mathbf{F}(\mathbf{x}, \mathbf{b}) + \epsilon \tag{17}$$

where $\epsilon$ includes the error of measurement and forward model. $\mathbf{F}(\mathbf{x}, \mathbf{b})$ depends on the retrieval quantity vector $\mathbf{x}$ (trace gas concentration profile) and on an additional parameter vector $\mathbf{b}$. The latter one includes quantities which have an impact on the measurement and are known with some accuracy (e.g. a priori information).

From the perspective of MAX-DOAS, this relationship is strongly ill-posed as the number of retrieval parameters is usually much higher than the number of measurements. Furthermore, as the light paths of two geometrically close elevation angles traverse similar vertical layers near the surface, measurements cannot be considered as independent. Consequently, the retrieval parameter vector $\mathbf{x}$ is not fully constrained by the input vector $\mathbf{y}$ which introduces the need for additional a priori knowledge which constrains the solution. With the assumption of a Gaussian error distribution, the optimal estimation method (OE) is often used to prevent unstable solutions when dealing with ill-posed problems (Rodgers, 2004).

$$\mathbf{x}_n = \mathbf{x}_0 + (\mathbf{K}^T \mathbf{S}_y^{-1} \mathbf{K} + g^{-1} \mathbf{S}_0^{-1})^{-1} \mathbf{K}^T \mathbf{S}_y^{-1} (\mathbf{y} - \mathbf{K}\mathbf{x}_0) \qquad \text{or}$$

$$= \mathbf{x}_0 + \mathbf{S}_0 \mathbf{K}^T (\mathbf{K}\mathbf{S}_0\mathbf{K}^T + g\mathbf{S}_y)^{-1} (\mathbf{y} - \mathbf{K}\mathbf{x}_0) \tag{18}$$

Again, $\mathbf{S}_0$ and $\mathbf{S}_y$ are the covariance matrices, $\mathbf{y}$ is the measurement and $\mathbf{x}_0$ is the a priori profile. Note that we introduced a scaling factor g which gives the possibility of regulating the weighting between a priori and measurement information. Here, the matrix $\mathbf{K}$ consists of BAMF values for every layer and measuring geometry (Eq. (5)) instead of weighting functions. The covariance matrix of measurements has only diagonal elements which are the absolute errors of the spectral DOAS fit of this absorber. $\mathbf{S}_0$ has constant variances on its diagonal elements but consists of additional non-diagonal elements based on a Gaussian distribution which accounts for a possible correlation of different profile layers (Barret et al., 2002).

## 4 Results and Discussion

In this section, we first present and discuss results of synthetic sensitivity studies. For this purpose, noise free differential slant columns densities of different aerosol and $NO_2$ scenarios were simulated with SCIATRAN. Subsequently, these datasets were used as input for BOREAS, and results are compared with the true profiles. In the next subsection, we discuss the error sources of the retrieval before, in the last section, results of data measured during the CINDI-2 campaign (2nd Cabauw Intercomparison of Nitrogen Dioxide measuring Instruments, 2016) are shown. These results are validated by comparison with different ancillary measurements (sondes, LIDAR, long path DOAS, ceilometer, AERONET, Pandora and in situ).





## 4.1 Sensitivity study with synthetic data

A synthetic dataset of differential slant column densities of $NO_2$ and $O_4$ was created using SCIATRAN's spherical mode including multiple scattering. The settings in Table 1 were chosen to describe conditions as they can be found in urban areas like the city of Bremen. Pressure and temperature profiles were taken from a U.S. standard atmosphere (NASA, 1976). The vertical

grid was set to 25 m steps from the surface up to 6 km, 250 m steps up to 10 km and 1 km steps up to 60 km. The LOS angles were chosen as a representation of a typical MAX-DOAS measurement with more angles for the lower elevations, where the sensitivity is highest. Both, solar zenith angle and relative azimuth angle were kept constant.The trace gas and aerosol profiles

**Table 1.** SCIATRAN settings for the calculation of differential slant column densities

| trace gases | wavelength (nm) | albedo | SZA (°) | RAA (°) | LOS (°) | asymmetry factor | SSA | climatology |
|---|---|---|---|---|---|---|---|---|
| $O_4$, $NO_2$ | 477 | 0.06 | 40 | 90 | 1, 2, 3, 4, 5, 6, 8, 15, 30 | 0.68 | 0.92 | US standard |

in Table 2 (see also Fig. 3, 6, A1 and A8) were chosen to assess BOREAS retrieval capability under simple (exponential profiles with scale height SH = 1 km) and non-ideal conditions (box profiles, which are not similar to the exponential a priori

profiles). The $NO_2$ scenarios did not include aerosols to avoid the mixture of uncertainties from the trace gas retrieval and inaccuracies within the aerosol retrieval. The box profiles are shown and shortly discussed in appendix A.

**Table 2.** Parameters for aerosol and $NO_2$ profiles used in the synthetic sensitivity study.

| | Exp. profiles (SH = 1 km) | | | Box profiles (appendix) | | |
|---|---|---|---|---|---|---|
| | E1 | E2 | E3 | B1 | B2 | B3 |
| AOT | 0.2 | 0.6 | 1.0 | 0.4 | 0.4 | 0.4 |
| VCD ($10^{15}$ molec/cm$^2$) | 5 | 10 | 20 | 10 | 10 | 10 |
| maximum altitude (km) | 6 | 6 | 6 | 0.5 | 1 | 2 |

### 4.1.1 Sensitivity of the aerosol retrieval

Figure 3 shows the true exponential extinction coefficient profiles (see Table 1) with aerosol optical thickness's of 0.2, 0.6 and 1.0. In addition, an a priori profile with the same scale height (SH = 1 km) but with AOT = 0.18 is depicted. In general,

an a priori profile should comprise known information of the true atmosphere. Ancillary measurements are frequently used to gather information for a mean profile. Here, we try to evaluate the performance of the algorithm by using an a priori profile close to the true profile (E1) but also for atmospheres with highly varying aerosol loads (E2, E3).





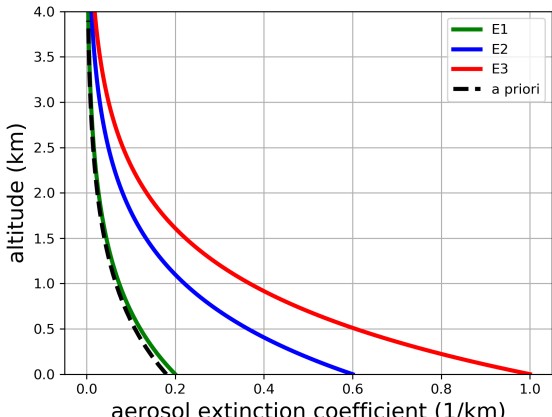

**Figure 3.** Exponential aerosol profiles for the synthetic sensitivity study. Also shown is the a priori profile for the aerosol retrieval (dashed line).

The retrieval uses vertical grid steps of 50 m for synthetic data because smaller step widths introduce retrieval noise, whereas larger steps result in vertical structure being overly smoothed. Since the retrieval tends to overestimate the upper layers due to missing sensitivity (see Fig. 9), a height depending a priori variance was used for the preparation of the a priori covariance matrix $S_a$. The measurement covariance matrix $S_e$ consists of a constant signal to noise (SNR) value which is multiplied with the measurement for each LOS to get absolute covariances. The Tikhonov parameter is changed in the first examples to demonstrate the retrieval response on this parameter. The iterations stop when convergence is reached or after 60 steps. Convergence in parameter space was set to $0.0001\,\mathrm{km}^{-1}$. In addition, convergence of the RMS of measured and simulated DSOT differences was set to 0.001.

On the left side in Figure 4 the retrieval results for different Tikhonov parameters $\gamma$ are shown. E1, which is close to the a priori profile (dashed grey line) is retrieved well for all $\gamma$ values. In general, smaller Tikhonov parameters give too much weight to the measurement by reducing smoothing and introducing retrieval noise which leads to oscillations. Large values smooth the solution in the direction of the a priori profile. When the true atmosphere is far away from the a priori profile, the algorithm does not perform satisfyingly. For E2, strong smoothing (large $\gamma$ values) leads to more or less straight lines in which insufficient curvature can be found, whereas weak (or no) smoothing means the beginning of oscillations indicated by the reduction of the bottom extinction. No parameter value is able to retrieve the proper bottom extinction. This can be found in the same way for E3 where all results show a clear lack in accuracy. This is not surprising as the a priori profile cannot be understood as the best guess or median of the atmosphere any more.

Several authors have already highlighted retrieval problems from MAX-DOAS measurements under highly variable (in time and space) atmospheric conditions (e.g. Vlemmix et al. (2011); Wagner et al. (2011); Vlemmix et al. (2015); Frieß et al. (2016)). Nevertheless, studies that focus on improvements of the regularization (Eriksson, 2000; Hartl and Wenig, 2013) or





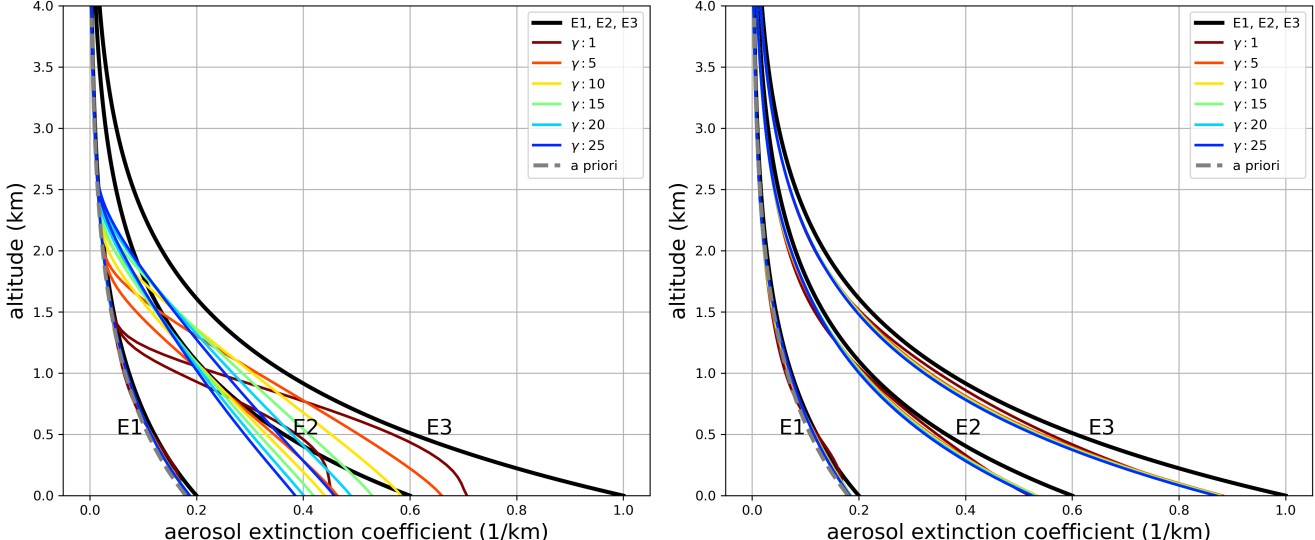

**Figure 4.** Retrieval results with a fixed (left) and a pre-scaled a priori profile (right) with varying Tikhonov parameters $\gamma$ for SNR = 3000. Small $\gamma$ values mean less smoothing of the resulting profiles.

a priori information (Frieß et al., 2006; Clémer et al., 2010; Hendrick et al., 2014) are rare, with respect to MAX-DOAS retrievals. Here, we use a priori pre-scaling to cope with strong deviations of the true atmosphere from the a priori estimate. For this purpose, the BOREAS aerosol retrieval is started on a reduced vertical and spectral grid from a zero profile (all a priori extinction values were set to be zero) as a priori information. Since the retrieval works directly on $O_4$ absorption, aerosol

optical thickness is in general the most reliable retrieval quantity. The AOT from this pre-calculation is used to scale the a priori profile for the true run. Note, that the only a priori information used both within the pre-run and within the main run is the Tikhonov parameter. This method performs well when the a priori shape is similar to the true atmospheric profile shape (right side of Fig. 4). However, no improvement is found when the shape of the a priori profile deviates strongly from the true atmospheric condition (see box profiles in appendix A). On the right-hand side of Figure 4, clear improvements can be seen

in both overall shape and bottom extinction. Nevertheless, the profiles tend to deviate stronger at the surface, especially when the aerosol load is higher which was also found by Frieß et al. (2006). The relative differences of retrieved AOT are high for the unscaled retrieval but can be significantly reduced with a priori pre-scaling (see Table 3). Further improvements could be made by increasing the allowed number of iterations together with stronger convergence criteria (not shown) but the average retrieval time would increase as well.

A comparison of all three profiles and their optimal $\gamma$ ranges with and without pre-scaling indicates that each combination of a priori and true profile has a specific optimal Tikhonov parameter which might differ strongly between each scenario. In addition, the signal to noise is another quantity which has a large impact on the retrieval. In general, SNR values might differ strongly throughout a day, because dynamic ranges of exposure and integration times are frequently used to operate the CCD





**Table 3.** True and retrieved AOT for SNR = 3000, $\gamma$ = 1.0 with and without a priori pre-scaling including relative differences to the true profile in brackets.

| AOT | true value | fix a priori | result | scaled a priori | result |
|-----|------------|--------------|--------|-----------------|--------|
| **E1** | 0.2 | 0.18 | 0.183 (-8.6%) | 0.161 | 0.179 (-10.4%) |
| **E2** | 0.6 | 0.18 | 0.439 (-26.9%) | 0.493 | 0.535 (-10.6%) |
| **E3** | 1.0 | 0.18 | 0.626 (-37.4%) | 0.847 | 0.895 (-11.1%) |

in the ideal saturation range when the number of photons might differ strongly (e.g. due to different viewing geometries). In Figure 5 we varied SNR (as a representation of the measurement error) and Tikhonov parameter $\gamma$ from 500 to 5000 and from 0.25 to 30 respectively, to find the best parameter range for the optimization of AOT and the bottom extinction (BOT). The figure shows results for the strong aerosol scenario E3 (E1, E2 and the box scenarios are shown in the appendix). Colour coded are the relative differences to the true value with bright spots indicating better retrievals. For scenario E3, high $\gamma$ (or low SNR)

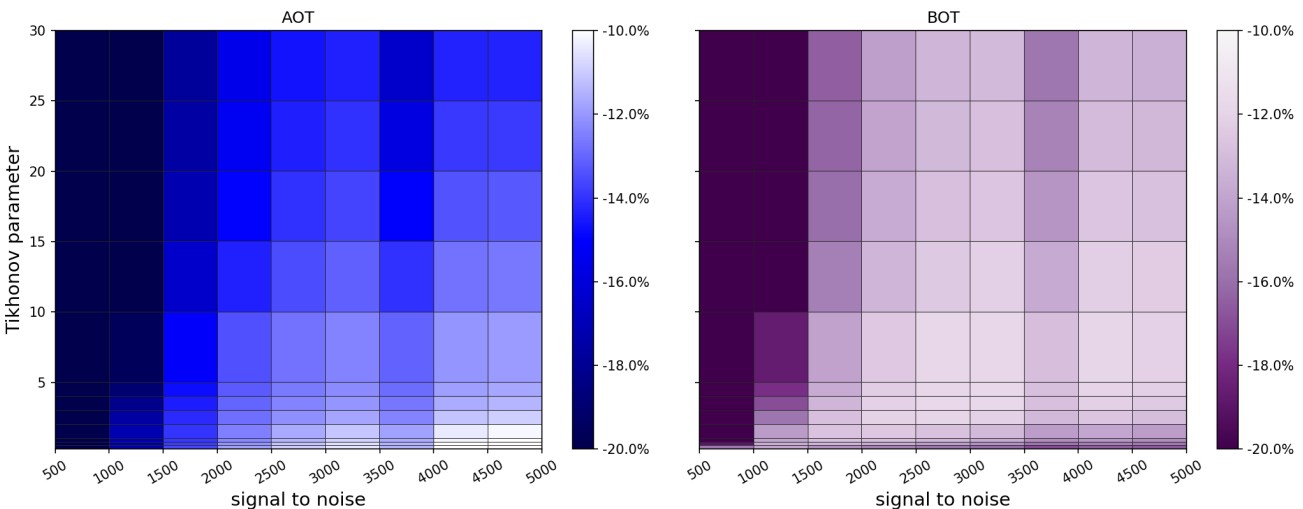

**Figure 5.** Variation of the relative difference of AOT and bottom extinction (BOT) to the true value as a function of signal to noise and Tikhonov parameter, colour coded for scenario E3 with a priori pre-scaling.

values apply too much smoothing (not enough weight on the measurement) which ends in strong AOT/BOT underestimations of the retrieval results. Furthermore, there is an ideal range of SNR values for $\gamma \leq 2$ where the AOT relative difference is $\leq 10\%$. In this specific SNR/$\gamma$ range, the bottom values do not show the best results. On the other hand, BOT values are optimized for $3 \leq \gamma \leq 10$ for SNR $= 2500$ and SNR $= 3000$ where the AOT does not have the lowest difference. As a consequence,

10 due to unequal vertical sensitivity, choice of a priori constraints and parametrization, in some cases either AOT or the shape

(c) Author(s) 2018. CC BY 4.0 License.





of the profile can be optimized. Stable solutions for the optimization of both profile characteristics could not be identified in the range of parametrization, regularization and convergence. As a result, a reasonable compromise for optimum values with respect to both AOT and BOT would be in the range SNR = 2000-3000 with low Tikhonov values $2 \leq \gamma \leq 5$. Note that the SNR range between 3500 and 4000 has larger relative deviations than the surrounding settings which could not be found for

other scenarios. This is assumed to be either a unique solution, where the range of settings leads to poor retrieval results, or a numerical instability.

In contrast to this pragmatic approach, the so-called L-curve approach (Hansen, 1992; Engl and Grever, 1994) was found to be insufficient as it already assumes a good regularisation, i.e. the data residual and the parameter norm ($\lVert \mathbf{x} - \mathbf{x_a} \rVert$) can be described well by a specific regularisation parameter (Hartl and Wenig, 2013), which was not found for the application of

MAX-DOAS profiling.

### 4.1.2   Sensitivity of the trace gas retrieval

The optimal estimation based trace gas profile retrieval faces similar problems as the regularization difficulties discussed for the aerosol retrieval in the previous section. In many studies, an a priori variance of a fixed percentage of the a priori profile in combination with the differential slant column density errors from the DOAS fit are used to constrain the solution (e.g. Frieß

et al. (2016)). Here, we show that this is an insufficient approach, since the fitting errors might be too small, giving too much weight to the measurement, when the $NO_2$ concentration is high and the a priori profile is not a good estimate of the true atmosphere.

Again, a priori pre-scaling is used to achieve a better estimate of the true atmospheric conditions. In contrast to the aerosol retrieval which needed a pre-run for a better first guess, the trace gas a priori pre-scaling is achieved using the 30° elevation

angle measurement. In the geometric approximation (see Eq. (4)), the differential AMF is 1 which leads to $\Delta S_{30} = V$. This vertical column density V is used as a pre-scaling value for the a priori profile.

The different $NO_2$ exponential profiles, based on the parameters in Table 2, are shown in Figure 6 together with the a priori profile (box profiles can be found in appendix A). Note that we used an a priori profile close to E2. These profiles are a good representation of the daily variability in an urban region, where high $NO_2$ concentrations are found during phases of commuter

traffic (rush hours) in the morning and late afternoon and low $NO_2$ loads in between.

Figure 7 depicts the profiling results without a priori pre-scaling on the left side and with pre-scaling on the right side. Different regularization ratios are achieved by varying parameter g in Eq. (18) which varies the measurement error by $\sqrt{g}$. The retrieval is done on the same vertical grid as in the previous section with similar variances for the a priori covariance matrix. In contrast to a constant signal to noise ratio, we use variances in the range of typical DOAS fit errors of $NO_2$ for the measurement

covariance matrix, which can be understood as the standard approach of the MAX-DOAS profiling community. For small elevation angles and high $NO_2$ concentrations, the relative $\Delta S$ error is often lower than $1\%$ and sometimes even lower than $0.3\%$, indicating a high reliability in the measurement but neglecting errors due to pointing inaccuracies of the telescope or horizontal inhomogeneities which affect the profiling result but not the measurement.

The results show again the importance of pre-scaling. Whereas the E2 scenario is well retrieved for all g-ranges without pre-

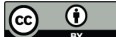



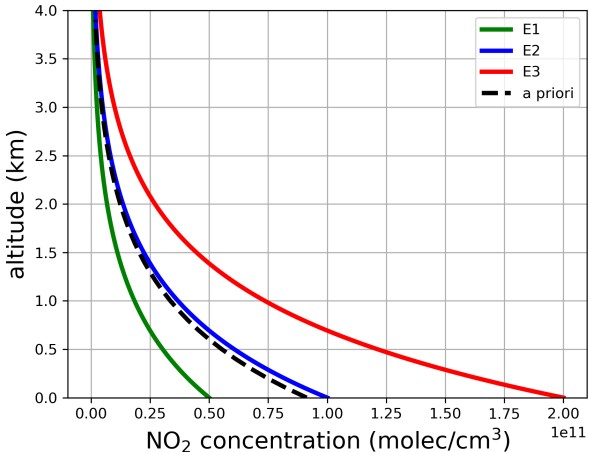

**Figure 6.** Exponential NO₂ profiles for the synthetic sensitivity study. Also shown, the a priori profile for the aerosol retrieval (dashed line).

scaling, the other scenarios tend to oscillate or underestimate the profile. Note, that the standard regularization ratio ($g = 1$) does not lead to satisfying results for E3. Applying pre-scaling leads to more accurate profiles especially for E3.

The specific optimal g-range for each scenario depends not only on the a priori profile and the true atmosphere but also on

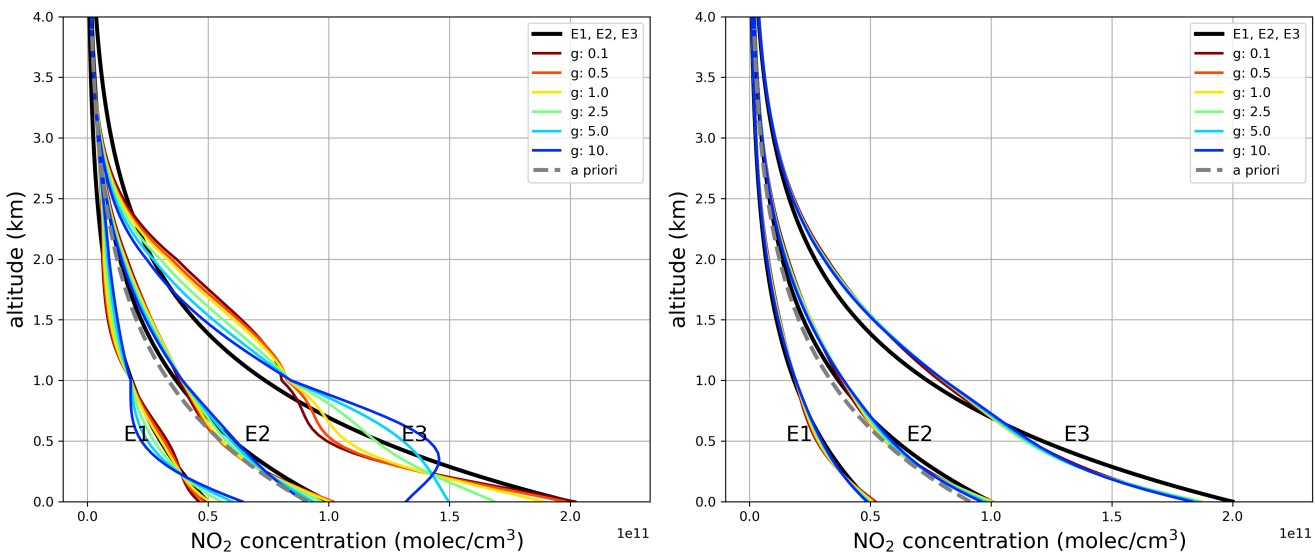

**Figure 7.** Retrieval results with a fix (left) and a pre-scaled a priori profile (right) with g factors. Small g values mean less measurement weighting for the resulting profiles.

the variance values, their height dependencies and the definition of off-diagonal elements. However, the quantification of this




optimal range is a difficult task for real data due to the lack of knowledge about the true atmosphere. Manual variations of g as well as iterative approaches are conceivable. Possible criteria for finding the best g values are e.g. the RMS between measurement and retrieval quantity or ancillary measurements. Here, we used fixed values which were chosen to decrease oscillations while maintaining the best surface concentration.

### 4.2 Discussion of error sources and information content

The total error of both aerosol and trace gas retrieval can be separated into three different error sources (Rodgers, 2004).

$$\mathbf{S}_{tot} = \mathbf{S}_{sm} + \mathbf{S}_{fw} + \mathbf{S}_{ns} \tag{19}$$

The first error term describes the smoothing error $\mathbf{S}_{sm} = (\mathbf{A} - \mathbf{I})\mathbf{S}_e(\mathbf{A} - \mathbf{I})^T$ which can be understood as error for the averaging kernel smoothed estimate of the true state of the atmosphere. $\mathbf{A}$ and $\mathbf{I}$ are the averaging kernel and identity matrix respectively. The second term of $\mathbf{S}_{tot}$ is the forward model error $\mathbf{S}_{fw}$, which is hard to quantify in real retrievals, because it needs the true state of the atmosphere. For near linear problems, this error source can be neglected, when the forward model parameters are well estimated. The last error is called retrieval noise $\mathbf{S}_{ns} = \mathbf{G}\mathbf{S}_e\mathbf{G}^T$ (G: gain matrix, see e.g. Eq. (16) and denotes the uncertainty of the retrieved profile due to measurement errors.

Figure 8 shows smoothing, retrieval and total error for the before mentioned scenarios of the aerosol retrieval (left) and the trace gas retrieval (right).

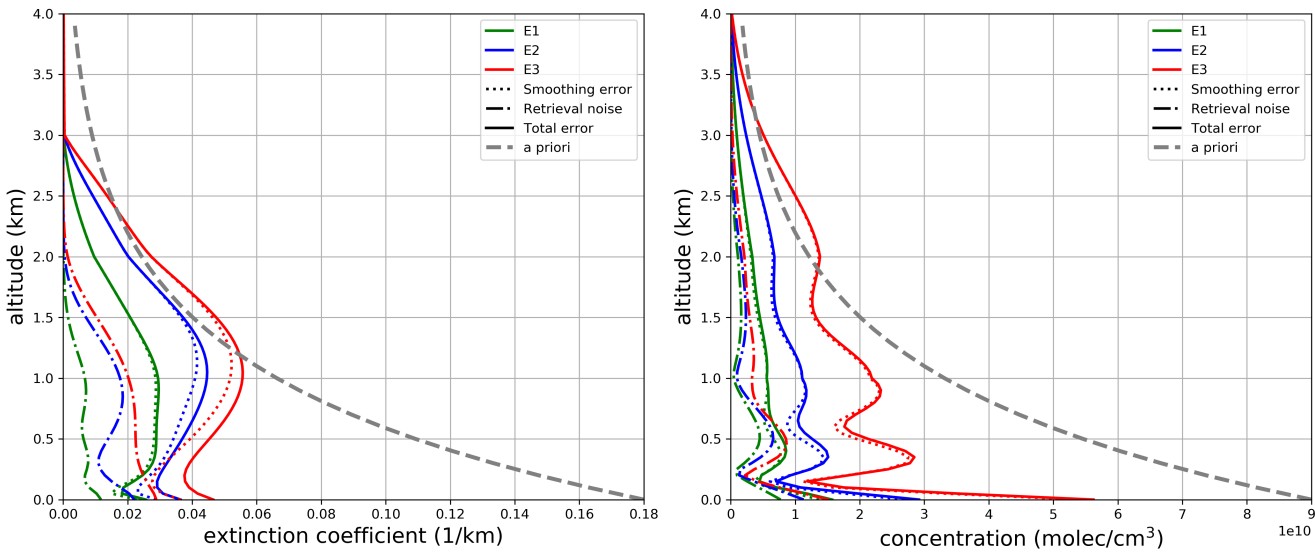

**Figure 8.** Retrieval errors for E1-E3 of the synthetic aerosol (left) and $NO_2$ study (right). Aerosol results are shown for SNR = 3000 and Tikhonov parameter $\gamma = 2$. Trace gas errors are depicted without additional regularization (g = 1). Pre- scaling was used.





The total errors for both aerosol and trace gas retrieval are dominated by the smoothing error with negligibly small retrieval noise. This shows, that the measurement itself can be considered as a small error source in contrast to the generally limited information content of MAX-DOAS measurements. Note, that the error components of the $NO_2$ profile show small oscillations which also indicates that larger constraints are needed. For the aerosol retrieval, the largest errors can be found in an altitude

were the sensitivity is low with an additional increase near the surface. In general, $NO_2$ shows also larger errors for higher altitudes but the surface error is dominant here, indicating a reduced sensitivity or poor parametrization. The errors for $z \geq 3$ are close to zero because the smallest a priori covariance is used as boundary condition for the highest altitudes, where the sensitivity is low.

For the quantification of the vertical sensitivity of the retrieval, Figure 9 shows the averaging kernels AK (see Eq. (16) and

text) for the aerosol scenarios E1-E3 calculated for a Tikhonov parameter $\gamma = 0.25$ and SNR = 3000. We used a small $\gamma$ to demonstrate the general vertical sensitivity rather than the smoothed one. Larger $\gamma$ values would lead to peaks at lower altitudes with slightly reduced peak sizes. Within an ideal retrieval, the AK matrix should be equal to the identity matrix (Rodgers, 2004). Then, delta peaks in every layer would indicate a perfect sensitivity of the retrieval to the true state of the atmosphere. For MAX-DOAS retrievals, AK are much broader with peaks close to the surface. Here, the sensitivity is highest and the

kernels for a certain layer show correlations with neighbouring layers due to the finite width of each kernel. The comparison of AK for E1-E3 reveals, that besides the surface kernel, all peaks are reduced in size and width for higher aerosol loads. This demonstrates the smaller sensitivity due to both, an insufficient a priori estimate and the additional aerosol load which reduces the general number of photons scattered into the light path (for specific geometries, e.g. RAA = $90°$). The area of the AK should ideally be 1 over a large range of altitudes. Here we find, that only within the first kilometer, the overall sensitivity is

high. Note, that especially for E2 and E3 the surface area value is much smaller than 1 due to the negative area of the lowermost kernels for higher altitudes. Negative kernel values indicate, that an additional aerosol load in this altitude would decrease the retrieved solution close to the surface. The FWHM of averaging kernels are a measure of the vertical retrieval resolution with the lowest values close to the surface. In the right hand side plot of Figure 9 we show FWHM at the nominal altitudes (coloured lines) and the altitudes of their individual maxima (black lines). The calculation was done by using the individual AK instead

of the FWHM of fitted Gaussian distributions. The limited vertical extent of the black curves indicate, that a theoretical vertical resolution with values between 500 m and 1 km is available for all kernel levels but the specific sensitivity for all layers lies within the lowest kilometer.

The trace of the averaging kernel matrix (degrees of freedom, DOF) is often discussed as the number of independent pieces of information for a retrieval. Hartl and Wenig (2013) already pointed out, that this number is not necessarily a strict limitation

of the retrieval and might differ strongly depending on the regularization.

In Figure 10, we depicted the RMS between measured and retrieved $O_4$ $\Delta S$ and the DOF for scenario E3. Giving the retrieval more freedom will increase the degrees of freedom and decrease the RMS which might be understood as more information used for a better measurement assimilation. However, in Section 4.3.1 we already demonstrated, that profiles with high DOF

and low RMS might oscillate. DOF only state the maximum number of independent pieces of information for the current



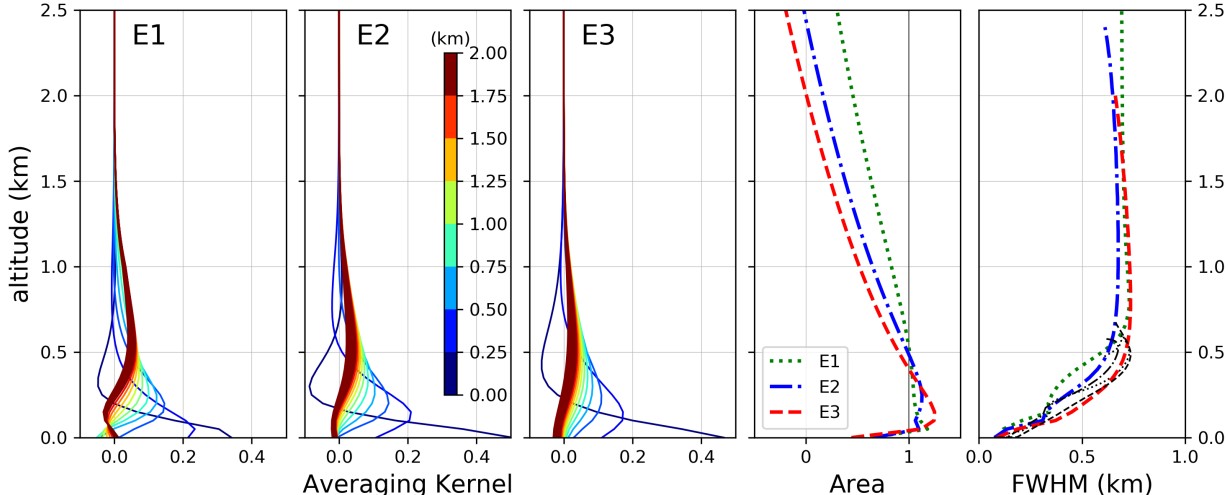

**Figure 9.** Averaging kernels of the aerosol retrievals for scenarios E1-E3 (first three figures on the left side). Also shown are the area and FWHM of the AK on the right side. FWHM are depicted on their nominal heights (coloured) and on the height of their individual peaks (black).

regularization parameters without considering if the retrieved profiles are optimal. Therefore, we do not use this quantity for the description of retrieval results.

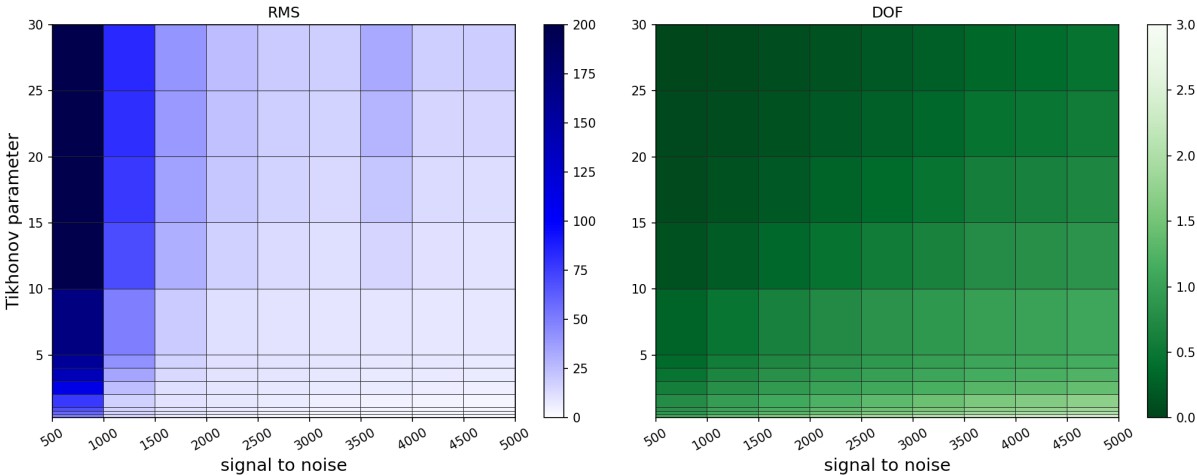

**Figure 10.** RMS between measured and simulated $O_4$ $\Delta S$ and the matching degrees of freedom (DOF) for the specific retrievals for different SNR and $\gamma$ values.



In Figure 11, we show averaging kernel, area and FWHM of the $NO_2$ retrieval for scenarios E1-E3. Similar conclusions can be drawn as for the aerosol retrieval for the shape and altitude variation of AK. The area is closer to 1 for all altitudes which is due to the missing smoothing of the Tikhonov term. The FWHM shows again a lower vertical sensitivity for higher layers. On the other hand, the FWHM curves at their specific peak heights (black) show a larger vertical spread in comparison

to the aerosol curves which indicates a larger sensitivity for higher altitudes in the $NO_2$ retrieval. The DOF for the trace gas retrieval lie in the range between 2.3 and 3.2 but decrease significantly when applying values of $g \neq 1$. In addition to the above

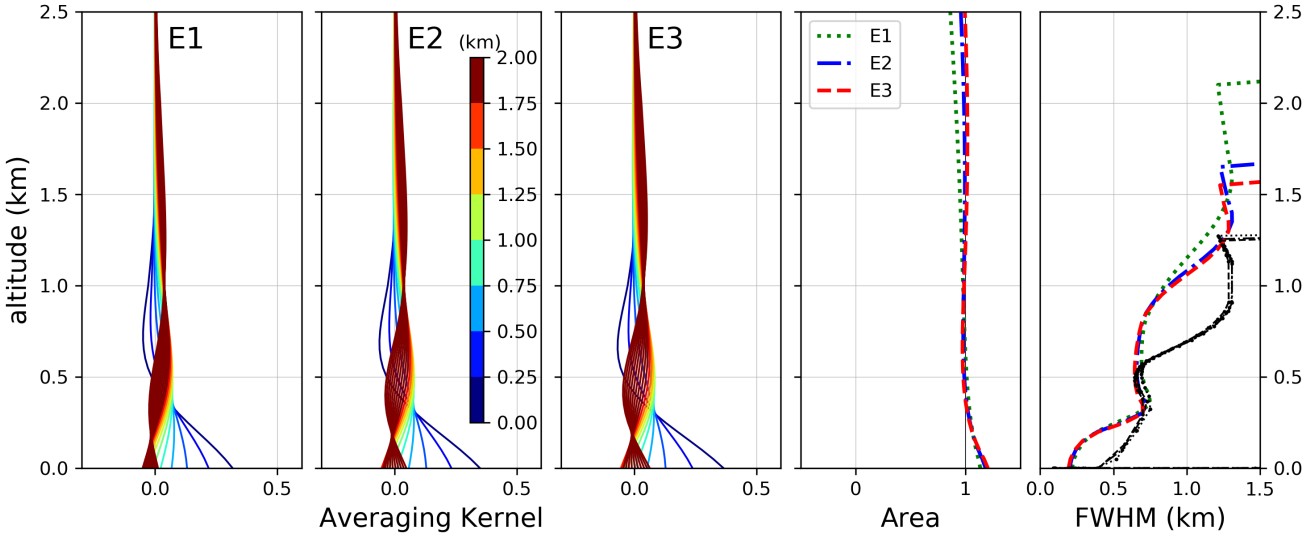

**Figure 11.** Averaging kernels of the $NO_2$ retrievals for scenarios E1-E3 (first three figures on the left side). Also shown are the area and FWHM of the AK on the right side. FWHM are depicted on their nominal heights (coloured) and on the height of their individual peaks (black).

mentioned errors and resolution properties, further error sources and problems should be noted. For real measurements, the true meteorological conditions are mostly unknown. Wang et al. (2017) pointed out, that wrong $O_4$ profiles might lead to strong deviations between retrieved and true aerosol profile. Furthermore, we did not implement the instruments field of view (FOV)

in the retrieval because it will increase run time. However, larger differences can be expected for concentrations/extinctions in the lowermost layers, when parts of the FOV might gather photons reflected on Earth's surface. Additionally, we assume a linearity between measurement and concentration which might be inaccurate for high aerosol loads and $NO_2$ concentrations when the telescope points along the horizon.

### 4.3 Retrieval of profiles during the CINDI2 campaign

The 2nd Cabauw Intercomparison of Nitrogen Dioxide Measuring Instruments campaign (CINDI-2) took place in Cabauw (the Netherlands) from 25.08 - 07.10.2016 and was funded by the European Space Agency (ESA) and the research groups participating. The campaign goals were the characterization of differences between $NO_2$ measurements by various instruments



and approaches, and the evaluation of the resulting datasets for the validation of the Copernicus satellite, Sentinel 5 Precursor (S5P). Over 40 different instruments operated by 30 groups from all over the world provided an outstanding ensemble of datasets for this task. The campaign was a successor of CINDI which was held in Cabauw from 16.06 - 24.06.2009 (see e.g. Piters et al. (2012); Pinardi et al. (2013); Zieger et al. (2011)).

5   The measurement site Cabauw is located in a rural region dominated by agriculture but is surrounded by four of the largest cities of the Netherlands (Rotterdam, Amsterdam, Den Haag and Utrecht). Thus, depending on the wind direction, long range transport from highly industrialized areas is likely which results in high pollution events. Here, three of the five investigated days show a more or less steady wind from south-easterly directions with a change in wind direction in the evening of the 15th of September (see Fig. 12). On the 23rd, wind came from the west whereas one day later the wind came mostly from southerly directions.

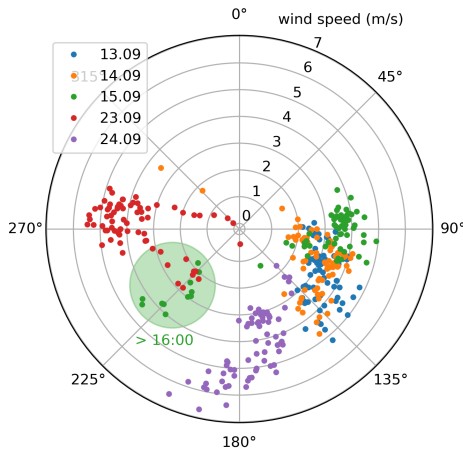

**Figure 12.** Wind speed and direction for the investigated days (CESAR, 2018) resampled to 5 minute values from 06:00 - 17:00 UTC.

In this study, the instrumental set-up for the validation of aerosol and trace gas profiles consists of in situ and remote sensing instruments. Near-surface concentration/extinction values are provided by ceilometer, $NO_2$-LIDAR, long path DOAS (LP DOAS) and in situ samplers (ICAD/CAPS, NAQMN). Integrated values are provided by Pandora and AERONET direct sun instruments. Besides the MAX-DOAS measurements, the only routinely retrieved profiling information comes from ceilometer

15   and $NO_2$-LIDAR data. In addition, three $NO_2$-sondes were launched on one of the days investigated here (15.09.2016). The profile validation is realized on two cloud free days (13. - 14.09.2016) and three days with broken clouds (15.09 and 23. - 24.09.2016). The operators of MAX-DOAS instruments were asked to perform elevation scans on the beginning of each hour and at 11:15 and 11:45 UTC every day. The ancillary measurements introduced in the following subsections were resampled or averaged on this time interval to prevent discrepancies due to time lags between two observations. Each scan took around

20   8 minutes and consisted of the following elevation angles: 1°, 2°, 3°, 4°, 5°, 6°, 8°, 15°, 30°. The telescope pointed into the western direction with a viewing azimuth angle of 107° from the south.





In this section, BOREAS retrievals were performed on data from the IUPB MAX-DOAS instrument on a 100 m step width vertical grid from the surface up to 4 km. The aerosol retrieval uses the a priori pre-scaling option with an exponential a priori profile described by a 1 km scale height and an aerosol optical thickness of 0.18. Asymmetry factor and single scattering albedo were chosen as 0.92 and 0.68, respectively. The SNR was set to 2500 with a Tikhonov-parameter of 2. The a priori variance was

decreasing with altitude from 1.5 at the surface to 0.01 at 4 km. The trace gas retrieval uses again a priori pre-scaling with an exponentially decreasing profile with a scale height of 1 km and a vertical column density of $9\times10^{15}$ molecules/cm$^2$. The a priori covariance matrix uses the same variances as the aerosol retrieval but includes Gaussian distributed side diagonal elements to account for correlations between individual layers (Barret et al., 2002). The measurement covariance matrix for $NO_2$ consists of the total differential slant column error on the diagonal elements only. Vertical profiles of pressure and temperature were

created by taking the mean of 16 different sonde measurements taken during the years 2013-2015 in De Bilt (the Netherlands).

### 4.3.1    Validation of aerosol retrievals

*Instrumentation for the aerosol validation*

Aerosol profiles were validated using three different instruments. The retrieved AOT was compared with values from an AE-RONET station (AErosol RObotic NETwork, Dubovik et al. (2000); Holben et al. (1998)). The Level 2 AOT at 440 nm was

scaled with the Ångström Exponent from the ratio 440 nm/675 nm to calculate the AOT at 477 nm (see Fig. 13). Due to measurement intervals varying between 4 and 30 minutes, we decided to resample the AERONET signal. The errors from AERONET instruments are usually in the range of 0.01 (VIS, IR) to 0.02 (UV) (Sayer et al., 2013). Here, we used a constant error of 0.01. The bottom extinction coefficient of the retrieved aerosol profiles was compared with in situ PM10 concentrations from the National Air Quality Monitoring Network (NAQMN) operated by the Dutch National Institute for Public Health and the Envi-

ronment (RIVM). The NAQMN measurement site Wielsekade is located at a distance of around 900 m from the MAX-DOAS station. The site is listed as a regional background station and the PM10 measurement principle is beta attenuation (Thermo Fisher Scientific FH62I-R). The instrument provides data on one minute intervals but only the hourly values are validated by RIVM. These hourly values were resampled on the MAX-DOAS times and the minute values were used to calculate the errors as standard deviation for the data points shown (see Fig. 14).

Furthermore, we used AERONET scaled ceilometer near surface extinction as a further validation dataset. The ceilometer (CHM15k Nimbus) was operated by the Royal Netherlands Meteorological Institute (KNMI) and sampled backscattering signals every hour at a wavelength of 1064 nm. The integrated backscattering signal was divided by the 1020 nm AERONET AOD to get a conversion factor which was applied to the backscattering signal for the conversion into extinction coefficients. This new ceilometer profile was again scaled with AERONETs AOT at 477 nm. The error is expressed as the standard deviation

calculated from the temporal and vertical averaging.

*Comparison of aerosol retrieval parameters*

Figure 13 shows the time series and the scatter plot of AERONET and BOREAS derived AOT. The BOREAS data was filtered for profiles which reached the iteration limit during the retrieval. Grey areas indicate clouds within the specified time period.





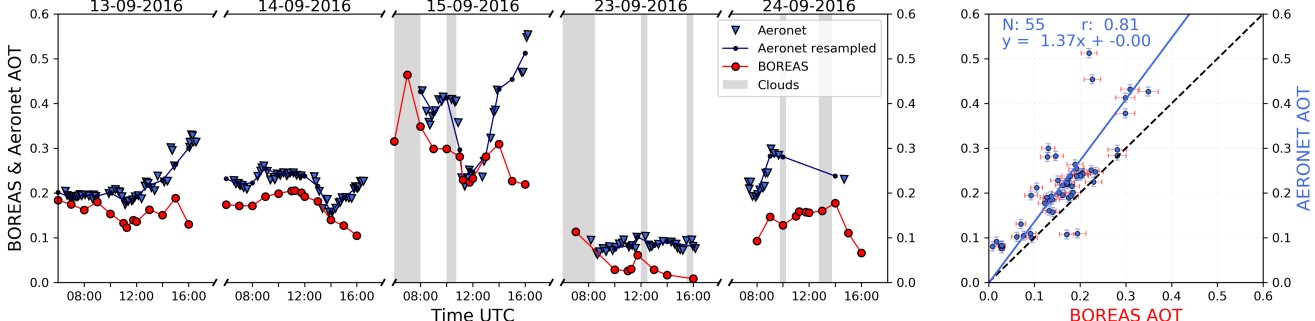

**Figure 13.** Left: time series of AERONET (blue) and BOREAS AOT (red). Small triangles show the original AERONET measurement, small dots with connecting lines depict the resampled data. Grey areas indicate clouds. Right: scatter plot of both datasets including parameters of the orthogonal regression and Pearsons correlation coefficient.

In the scatter plot, BOREAS errorbars were calculated by integrating the total error (Eq. (19)) vertically. On the first two cloud free days, the temporal variability shows a similar pattern with a slight offset between both instruments most likely introduced by an underestimation of the AOT by BOREAS. Due to the limited sensitivity of MAX-DOAS profiling for higher altitudes (see Fig. 9), elevated aerosols do not contribute to BOREAS AOT but can be measured by direct sun instruments like AERO-

NET sunphotometers.

In the evening, results of both instruments vary more. Different reasons for this finding are possible. First, a developing planetary boundary layer (PBL) might exceed the vertical extension where BOREAS has sufficient sensitivity. Second, when pointing towards the sun, saturation of the CCD becomes a problem leading to low integration times which decrease SNR and fitting quality. In addition, RTM calculations might introduce uncertainties at high SZA, when the aerosol load is high and

the light path is strongly increased. Furthermore, the aerosol phase function used leads to uncertainties as the forward scattering peak might be underestimated by the Henyey-Greenstein parametrization. The other days show a higher variability due to an increase in cloudy scenes. Note that especially in the morning and around noon, clouds may influence the instruments in various ways because of their different azimuthal viewing directions. BOREAS profile retrieval cannot be considered as reliable when one or several elevation angles pointed into clouds during the measurements, as intensity and light path vary in

unpredictable ways. This might also lead to large iteration numbers or to no convergence at all. On the 15th of September, the temporal patterns of the two instruments differ strongly. Although the extinction values around noon agree well, AERONETs decrease before and the increase afterwards cannot be found this strongly by BOREAS. Especially in the evening, both curves deviate, indicating highly variable atmospheric conditions which might also be introduced by a change in wind direction (see Fig. 12). The aerosol load on the last two days is much lower because of rainfalls in the time between 16.09 and 22.09. In the

morning of the 23rd, several profiles were discarded due to exceeded iteration limits.

The correlation coefficient of 0.81 is high and shows the general good agreement at three of five days. The regression line indicates the before mentioned small underestimation of the AOT by BOREAS.





Figure 14 shows the comparison of near-surface aerosol values. Extinction coefficients from BOREAS and ceilometer are depicted together with PM10 concentrations of the NAQMN in situ instrument. Note that the y-axis for PM10 concentrations was chosen to match the results of BOREAS on the 14th of September. The NAQMN in situ sampler and BOREAS show a

good temporal agreement within the first three days. Only in the morning hours at the 13th and 15th, data from the two instruments show a larger spread. This deviation is most likely due to the insufficient vertical resolution of MAX-DOAS profile retrievals when the PBL has not yet evolved and near surface aerosol loads are dominant (see Fig. 9 and 15 for further details). On the 15th, early morning clouds might also have an impact on the deviation. Again, a larger variability for BOREAS results can be found for the last two days, when the retrieval was influenced by broken clouds. The in situ instruments show a smoother

daily variation here. The ceilometer bottom extinction shows a good agreement with BOREAS and NAQMN on the first day and an underestimation on the second day. On the 15th of September, bottom values are influenced by thick clouds which might interfere with the backscattering signal of altitude ranges below the clouds (see Fig. 16). The 23rd shows more variability than the first two days but this can be found for the other instruments as well.

The correlation of BOREAS bottom extinctions with data from both validation instruments is high, indicating the good

agreement on cloud free days. The correlation with ceilometer near-surface values is high as well but the regression line shows a general underestimation of ceilometer near surface values in comparison to BOREAS.

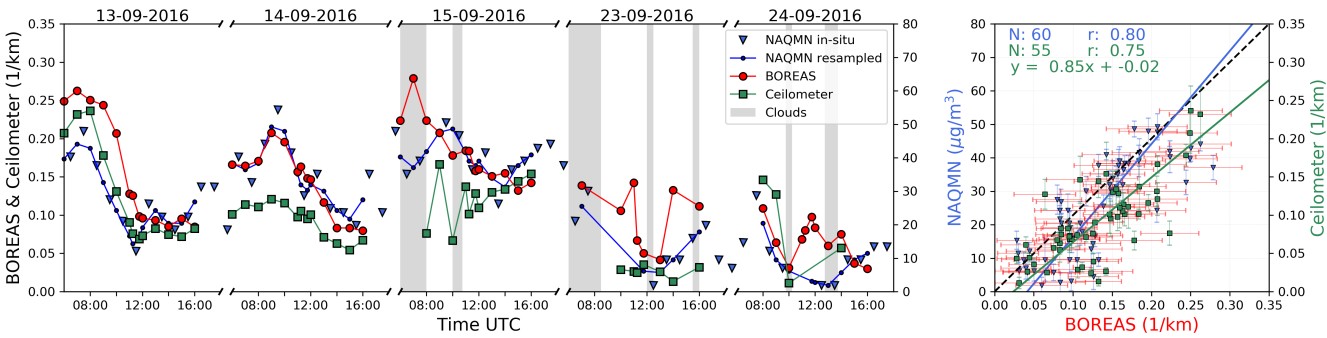

**Figure 14.** Left: time series of BOREAS (red), ceilometer (green) and NAQMN in situ (blue) and near surface aerosol parameters. Small triangles show the original hourly NAQMN measurements, small dots with connecting lines depict the resampled data. Green squares show ceilometer near-surface extinction values evaluated by averaging the 10 - 50 m coefficients. Grey areas indicate clouds. Right: scatter plot including parameters of the orthogonal regression and Pearsons correlation coefficients.

Figure 15 shows the aerosol layer height found by the ceilometer for all days. Especially during the first three days, an upward extending PBL can be identified from noon to the late afternoon. Only on the 15th, a more or less stable residual PBL seems to exist in the morning hours. The other days show a high variability in the morning with layer heights from 200 m to

600 m with more or less individual high signals which might be produced by smaller clouds at higher altitudes. The 24th of September shows a stable layer around 200 m and a diffuse developing PBL in the afternoon.





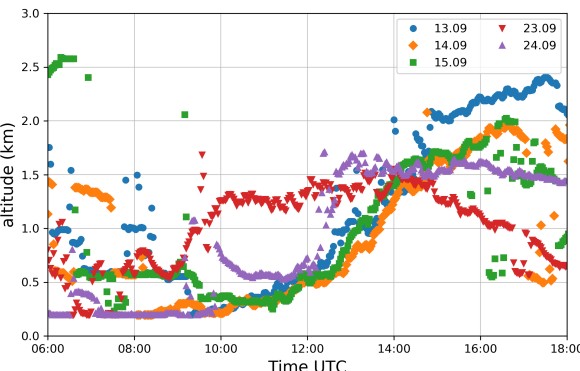

**Figure 15.** ceilometer aerosol layer height within the planetary boundary layer.

The underlying extinction coefficient profiles for the above discussed bottom extinction and AOT values can be seen in Figure 16. In the top row, temporal and vertically averaged ceilometer profiles are depicted. In the mid row, this data was smoothed to the MAX-DOAS vertical resolution by the application of BOREAS averaging kernels ($\mathbf{x}_{new} = \mathbf{x}_{apri} + \mathbf{A}(\mathbf{x}_{ceilo} - \mathbf{x}_{apri})$ see Rodgers and Connor (2003)). The first day shows a good agreement of BOREAS and ceilometer near surface extinction with a small offset between both curves. In the morning, both instruments find the aerosol load mainly located in the lowermost layers. Beginning around noon, the PBL starts developing with a maximum height of 2 km in the afternoon found by the ceilometer. BOREAS can not resolve this increasing PBL as well as the averaged ceilometer but the AK smoothed data indicate that the reason might be a limited sensitivity for the top boundary of the PBL.

On the 15th of September, the averaged ceilometer data show high and thick clouds in the morning and a rising PBL in the afternoon. In the AK smoothed data, these clouds can not be identified anymore. BOREAS introduces elevated aerosol layers which can be understood as a retrieval artefact due to these cloudy scenes. In the afternoon, BOREAS finds an upward expansion in the PBL similar to the ceilometer with the exception of the last profile which was influenced by the telescope pointing towards the sun. BOREAS extinction values are smaller in the PBL which is a consequence of the already explained underestimation of BOREAS AOT to AERONETs AOT which was used for the backscatter signal scaling.

### 4.3.2 Validation of nitrogen dioxide profiles

***Instrumentation for the $NO_2$ validation***

Several instruments were used for the validation of BOREAS nitrogen dioxide profiling results. The VCD results were validated with the help of a Pandora instrument (#128) operated by LuftBlick (Herman et al., 2009) and a $NO_2$ LIDAR from RIVM. In addition, three $NO_2$ sondes launched on the 15th of September by KNMI were also used.

The Pandora instrument was used with its direct sun capability to retrieve total nitrogen dioxide columns on hourly timesteps (see Fig. 17). Since the measurement time did not match to BOREAS scans, Pandora values were resampled. The stratospheric





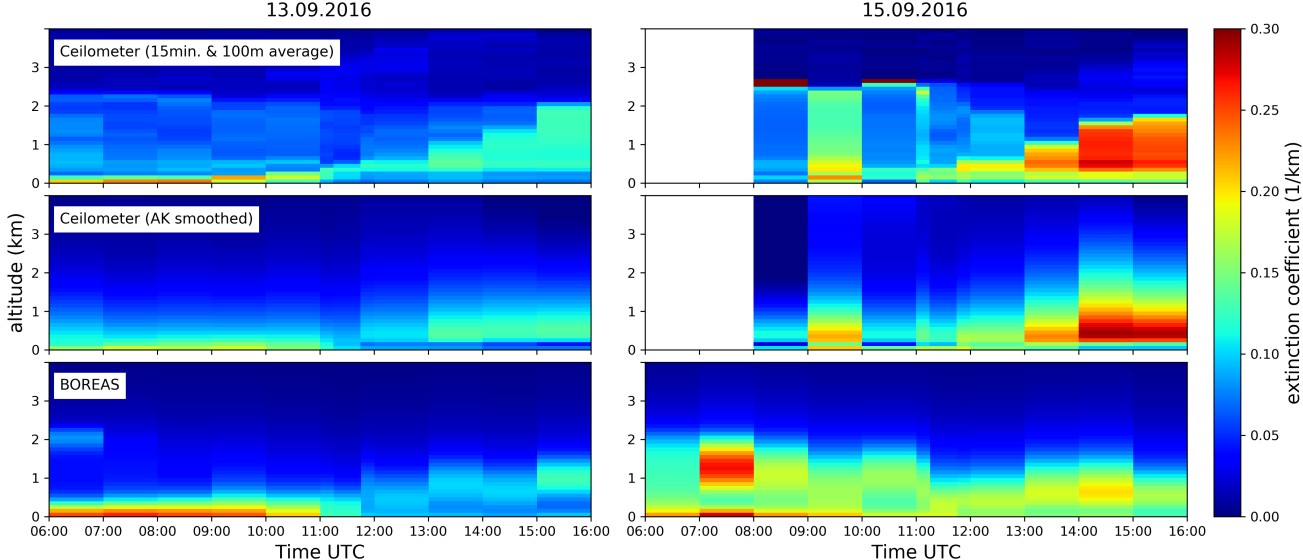

**Figure 16.** Comparison of aerosol extinction coefficient profiles from BOREAS and ceilometer for the 13.09.2016 (left) and 15.09.2016 (right).

column was subtracted based on the approach of Knepp et al. (2015) by using OMI stratospheric columns during the Cabauw overpass (L2 OVP NASA (2018)). The data was quality filtered and the errors were provided by LuftBlick and are based on Herman et al. (2009).

The $NO_2$ LIDAR (Brinksma et al., 2008) measured not as frequently as Pandora and MAX-DOAS instruments but performed several measurements on the 15th of September. One LIDAR scan took approximately 30 minutes with the temporal midpoints of the scans on a non-regular temporal grid which necessitated resampling. The scan was done on different elevation angles which enabled the retrieval of vertical profiles with altitude points up to 2.5 km. For the VCD calculation, these profiles were vertically integrated. Errors were provided by the RIVM team and Gaussian error propagation was applied for the calculation of VCD errors. In addition to the remote sensing instrument, $NO_2$ sondes were launched on the 15th around 5:15, 8:04 and 10:25 UTC. The sondes provide $NO_2$ VMR as well as pressure, temperature and altitude data for the conversion to concentrations which were vertically integrated up to 4 km.

Near surface concentration validation was done with the help of in situ (NAQMN, ICAD, CAPS) and remote sensing instruments (LIDAR and LP DOAS). The in situ samplers NAQMN, CAPS and ICAD were operated by RIVM, the Royal Belgian Institute for Space Aeronomy (BIRA-IASB) and the institute for environmental physics in Heidelberg (IUPH), respectively. In addition to the aerosol bottom concentration, NAQMN provided also $NO_2$ concentrations (Teledyne API 200E, see Section 4.3.1 for details on data handling). IUPH and BIRA-IASB created a common $NO_2$ in situ dataset from IUPHs ICAD (Iterative CAvity DOAS, (Pöhler et al., 2017)) and BIRAs CAPS (Environment SA, AS32M- CAPS, Cavity Attenuated Phase Shift, (Kebabian et al., 2008)). This joint dataset filled gaps in the individual measurement series and fixed a scaling





issue for CAPS. BIRA-IASB operated two different CAPS instruments which were installed at the 27 m and 200 m level of the Cabauw measurement tower. The joint ICAD/CAPS dataset as well as the 200 m CAPS data were averaged over 15 minutes time steps. For both instruments (ICAD/CAPS) an error of 1 ppb was assumed.

In addition to the in situ instruments, a long path DOAS (LP DOAS) provided by IUPH (Pöhler et al., 2010) was used for the

validation. Four reflectors attached on different altitude levels of the measurement tower (12.7 m, 47 m, 107 m and 207 m) provided unique and well-defined light paths which enabled the calculation of concentration on several altitudes. The instruments elevation measurements lead to profiles every 30 minutes. Errors were calculated by the IUPH team.

Nitrogen dioxide profiles were provided only by the $NO_2$ LIDAR and sonde launches. The LP DOAS did not cover enough altitude levels for a comparison. The conversion from trace gas VMR to concentrations for all datasets was done with the

meteorological data from the CESAR observatory (CESAR, 2018).

*Comparison of $NO_2$ retrieval parameters*

Figure 17 depicts the comparison of VCD for different instruments. The agreement with Pandora is good on all days with larger deviations in the morning and on the 15th of September in general with the exception around noon. Deviations in the

morning might be (similar to the AERONET comparison) due to spatial differences in the $NO_2$ distribution, when Pandora points to the east and MAX-DOAS to the west. This argument is supported by the LIDAR measurement on the 15th that agreed well with BOREAS and shared the same azimuthal viewing direction. In the afternoon of that day, BOREAS VCD increase faster than Pandora and LIDAR with a strong overestimation for the last profile at 16:00 when the sun was low. Here, we can exclude horizontal gradients as a reason for differences, because of the same viewing direction of LIDAR and MAX-DOAS.

On the same day, $NO_2$ sondes proved the BOREAS VCD only during the ascend (triangle with edge to the top) of the second launch correct. The matching descend (triangle with edge to the bottom) for the first flight was approximately 38 km into the northwestern direction between Amsterdam and Rotterdam and agrees more with Pandora columns indicating differences due to azimuthal viewing directions. The strong differences in VCD for ascend and descend of the first and second sonde launches show a high temporal and spatial variability of the $NO_2$ concentration which is supported by all remote sensing instruments.

The correlation with Pandora is slightly better than with LIDAR but the regression parameter shows, that Pandora VCDs are higher than those from BOREAS whereas the LIDAR regression is closer to one.

In Figure 18, near surface $NO_2$ concentrations are depicted. Both in situ datasets (NAQMN and ICAD/CAPS) agree well with each other and show larger differences only in the morning of the 14th of September. The LP DOAS profiles at 12.7 m and 47 m show similar concentrations as the in situ instruments. Differences in the morning of the first two days between the lower and

the higher LP DOAS values indicate a strong vertical inhomogeneity which was also found for the aerosols (see Section 4.3.1). BOREAS surface concentrations agree very well with all datasets except during the morning hours. On the 13th of September, in addition to the ICAD/CAPS data at the 27 m level (blue), the 200 m points are shown (green). The anti-correlated behaviour in the morning hours confirms the strong inhomogeneity and proves that nitrogen dioxide was mainly concentrated close to the surface at that time. The mean values of both CAPS instruments (yellow line) show a better agreement with BOREAS than the

27 m and 200 m data.



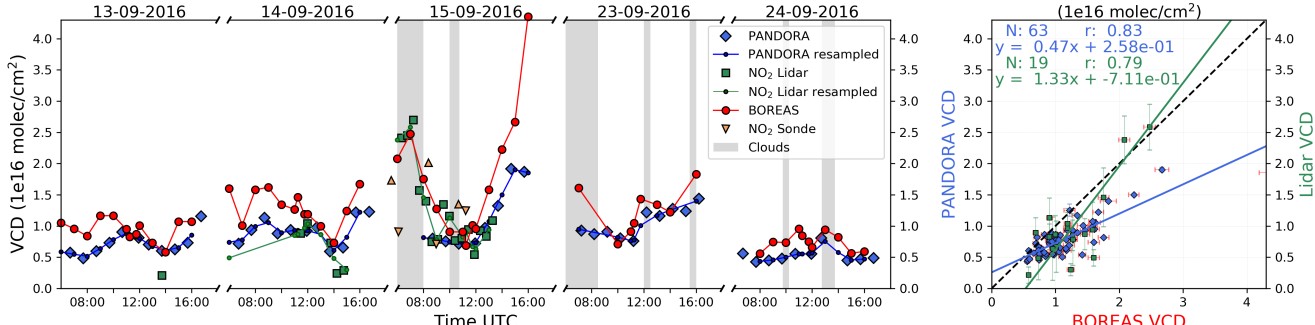

**Figure 17.** Left: time series of Pandora, LIDAR and BOREAS NO$_2$-VCD. Blue diamonds show the original Pandora measurements, green squares the LIDAR datapoints. Small dots with connecting lines depict the resampled datasets of Pandora and LIDAR. BOREAS is shown as red circles. Grey areas indicate clouds. The plot includes orange triangles as the representation of the integrated NO$_2$ sondes measurements with the ascend and descend separated in different triangles with the edge to the top or to the bottom respectively. Right: scatter plot of the datasets including parameters of the orthogonal regression and Pearsons correlation coefficients.

It seems that BOREAS cannot fully resolve this thin near-surface layer and retrieves rather smooth profiles instead of sharp concentration peaks (see also the discussion about vertical sensitivity in subsection 4.2). This is again in agreement with LP DOAS results on the 24th of September, where the 200 m level LP DOAS finds similar concentrations as BOREAS. Note, that the NO$_2$ LIDAR has a finer resolution within the lowest 100 m which might be the reason, why it depicts similar concentration

numbers as the other instruments in the morning hours of the 15th of September. As a rough estimation of the thickness of the individual thin layers in the morning, we divided BOREAS VCD by the ICAD/CAPS concentration and found, that the layer height lies around 200 m (06:00 - 08:00), 300 m (09:00) and 400 m (10:00) on the 13th of September. The mean AK FWHM for the surface layer on that day is 214 m indicating that the lowermost layer can be resolved only from around 09:00 UTC where the curves approach each other. A good agreement is found at 10:00 UTC when the concentration is focused in a layer

twice as thick as BOREAS surface resolution.

The correlations are high for all instruments with the highest value for NAQMN and the NO$_2$ LIDAR. The in situ instruments show similar correlations but a slight underestimation of BOREAS can be found with slopes between 1.24 and 1.45.

Figure 19 depicts NO$_2$ profiles for BOREAS, LIDAR and sondes on the 15th of September. Similar to the aerosol profile comparison, averaging kernels were applied to LIDAR and sondes measurements. The vertical profiles of the unsmoothed

LIDAR show the previously discussed high vertical inhomogeneity. We can clearly identify a high concentration close to the surface and another elevated NO$_2$ layer with altitudes varying between 100 m and 500 m. The unsmoothed sondes agree well with the LIDAR measurements only at the ascend of the second launch. After the application of averaging kernels, the two distinct layers with high concentrations are smoothed to one layer with NO$_2$ concentrated at the surface indicating again a lower vertical sensitivity of MAX-DOAS profiling. In addition, smoothed and unsmoothed LIDAR find larger concentrations

for altitudes over 500 m before 8:00 UTC which was not found by BOREAS and the sonde profiles.



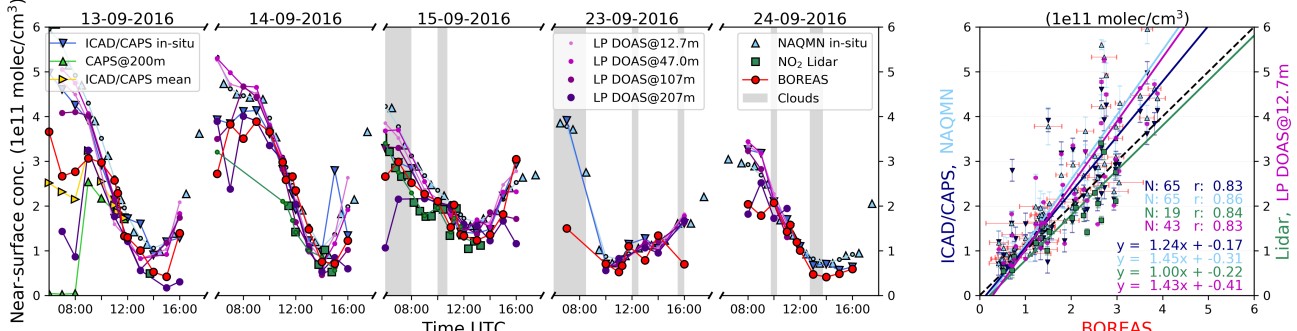

**Figure 18.** Left: time series of in situ and remote sensing NO$_2$ near surface concentrations. Triangles show in situ instruments for NAQMN (bright blue, triangles pointing upwards) and for ICAD/CAPS (dark blue, triangles pointing downwards). In addition, on the 13.09.2016, CAPS at the 200 m level is depicted as green triangles with the edge up and the mean values of both CAPS with the edge to the side (yellow). The NO$_2$ LIDAR is plotted as green squares and the LP DOAS as different sized circles in shades of magenta (lowest altitude as smallest circle). BOREAS is shown as large red circles. Right: scatter plot of the datasets including parameters of the orthogonal regression and Pearsons correlation coefficients.

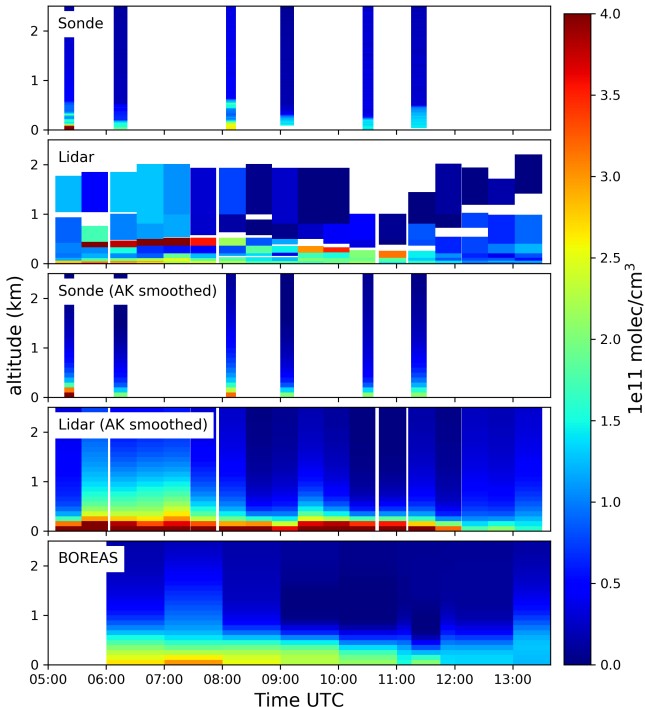

**Figure 19.** Comparison of NO$_2$ concentration profiles from BOREAS and LIDAR and sonde measurements for the 15.09.2016.





## 5  Conclusions

In this study, we introduced BOREAS, a new profiling algorithm for MAX-DOAS measurements. BOREAS retrieves aerosol extinction coefficient profiles around $O_4$ absorption bands from the oxygen dimers differential slant column densities. In contrast to existing inversion algorithms which are based on perturbation theory, BOREAS directly minimises the difference of

modelled and measured optical depth by varying aerosol extinction coefficient profiles in an iterative Tikhonov regularization scheme. In addition, trace gas concentration profiles are retrieved via optimal estimation using the aerosol properties determined in the first step. For both retrieval parts, we introduced a priori pre-scaling as a simple method for improving results under variable atmospheric conditions.

The retrieval performance was demonstrated on the example of synthetic data calculated with SCIATRAN and with real mea-

surements performed during the CINDI-2 campaign in Cabauw 2016. The synthetic scenarios were chosen to work as a stress test for BOREAS. One fix a priori profile was used for the retrieval of different large aerosol loads and $NO_2$ concentrations. The results are good when the scenario is relatively close to the a priori profile but fail for strong loads/concentrations. In a comprehensive variation of parameters we showed that, in non-ideal situations, either the shape/near-surface value or the integrated quantity can be optimized. The information content of a MAX-DOAS scan with degrees of freedom between $\approx 1$

and $\approx 3.5$ is not enough for a perfect retrieval. Furthermore, the frequently utilized assumption of using the trace gas errors from DOAS fits only as measurement variance was found to be insufficient since the regularisation ratio might be wrongly weighted. We suggest a manual change in measurement errors or an iterative approach for finding the best ratio.

In the second part of this study, BOREAS retrieval results from measurements of the IUPB MAX-DOAS instrument during the CINDI-2 campaign were validated with ancillary measurements. The agreement for all parameters was good with correlations

equal or higher than 0.75. Systematic offsets or deviations depending on geometry, atmospheric condition and investigated air mass indicate limitations due to the specific measurement characteristics which were also found in earlier MAX-DOAS profiling studies. The correlation with AERONET AOT is 0.81 with an underestimation by BOREAS leading to a regression slope of 1.37. The bottom value correlation is 0.75 for the ceilometer near surface extinction and 0.8 for NAQMN in situ measurements. The total column of $NO_2$ shows a good agreement with Pandora and LIDAR with correlations of 0.83 and 0.79

respectively. In contrast to the underestimation of BOREAS integrated aerosol, Pandora nitrogen dioxide columns are overestimated, especially in the morning hours. The correlations for near surface $NO_2$ concentrations are high with values larger than 0.83. Stronger deviations between BOREAS and in situ were found in the morning. This was explained by the limited vertical resolution of BOREAS, while in the morning the highest concentrations are found close to the surface.

As a conclusion, BOREAS profiling capabilities are strong which was proven by high correlations with all validating instru-

ments. Discrepancies were found due to different azimuthal viewing directions of the instruments and the limited vertical resolution of MAX-DOAS profiling when aerosol load or concentration is close to the surface in layers thinner than the BOREAS vertical resolution. Comparison of BOREAS results to the performance of other MAX-DOAS profiling algorithms will be discussed in two upcoming papers (Frieß et al. (2018) and Tirpitz et al. (2018)).





*Acknowledgements.* This study was supported by the University of Bremen and by the FP7 Project Quality Assurance for Essential Climate Variables (QA4ECV), no. 607405. Participation of the University of Bremen in the CINDI-2 campaign received funding from ESA via RFQ/3-14594/16/I-SBo. The authors would like to thank the CINDI-2 team for excellent support on site. Furthermore, we thank François Hendrick and Marc Allaart for the provision of mean pressure and temperature profiles used within the BOREAS retrieval of CINDI2 data

**Appendix A: Additional plots and retrieval of box profiles**

**A1    Additional aerosol scenarios and plots**

The retrieval results for the box scenarios in Figure A1 are shown in Figure A2 for different Tikhonov parameters and SNR = 3000. A priori pre-scaling leads only for scenario B3 to a clear improvement since the bottom extinction is reached. For B2 and B3, the resulting profile shapes change slightly which indicates the before mentioned lack in improvements through

pre-scaling, when the a priori profile shape differs strongly from the true atmosphere. Nevertheless, since exponential profiles are improved a lot and box profiles show no degradation we highly recommend the usage of better and more flexible a priori profiles.

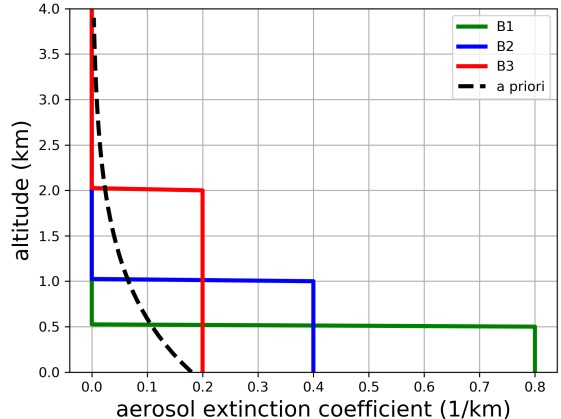

**Figure A1.** Box aerosol profiles for the synthetic sensitivity study. Also shown, the a priori profile for the aerosol retrieval (dashed line).

Figures A3 till  A5 show the SNR and Tikhonov variation for the three aerosol box profiles. One can see, that for B2 the

smallest $\gamma$ values lead to the best bottom value, whereas for scenario B1 these values would result in an overestimation. Also the optimal SNR/$\gamma$ range for the AOT differs strongly for all box profiles which show again, that regularization should be considered as a flexible choice of parameters, depending on the specific atmospheric conditions.





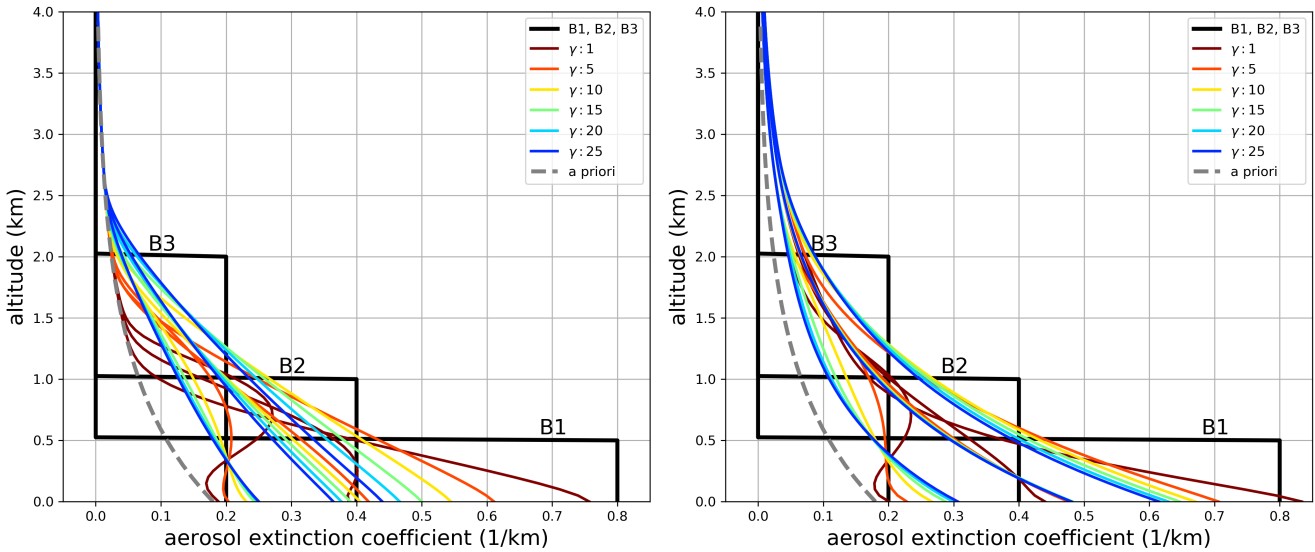

**Figure A2.** Retrieval results with a fix (left) and a pre-scaled a priori profile (right) with varying Tikhonov parameters $\gamma$ for SNR = 3000. Small $\gamma$ values mean less smoothing of the resulting profiles.

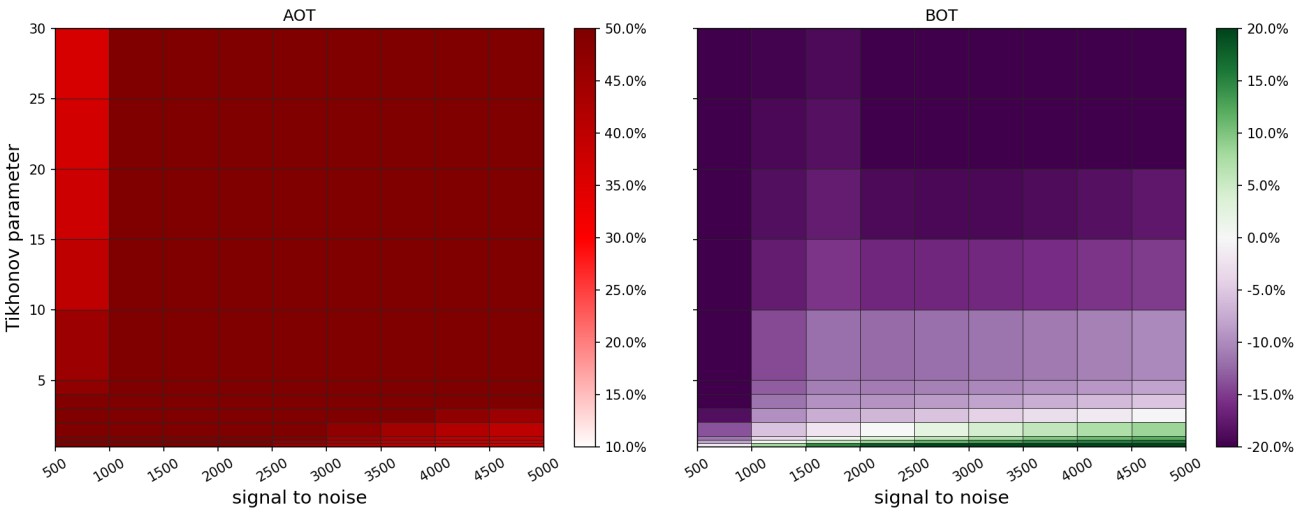

**Figure A3.** Variation of signal to noise and Tikhonov parameter with the relative difference of AOT and bottom extinction to the true value, colour coded for scenario B1 with a priori pre-scaling.

Figures A6 and A7 show the SNR/Tikhonov variation for the exponential profiles E1 and E2. Note that for E1 all parameter combinations lead to good profiling results because the a priori profile is close to the true atmosphere. In comparison to E3, E2 shows a better adaptation of AOT with a coincidently similar performance for the bottom extinction.





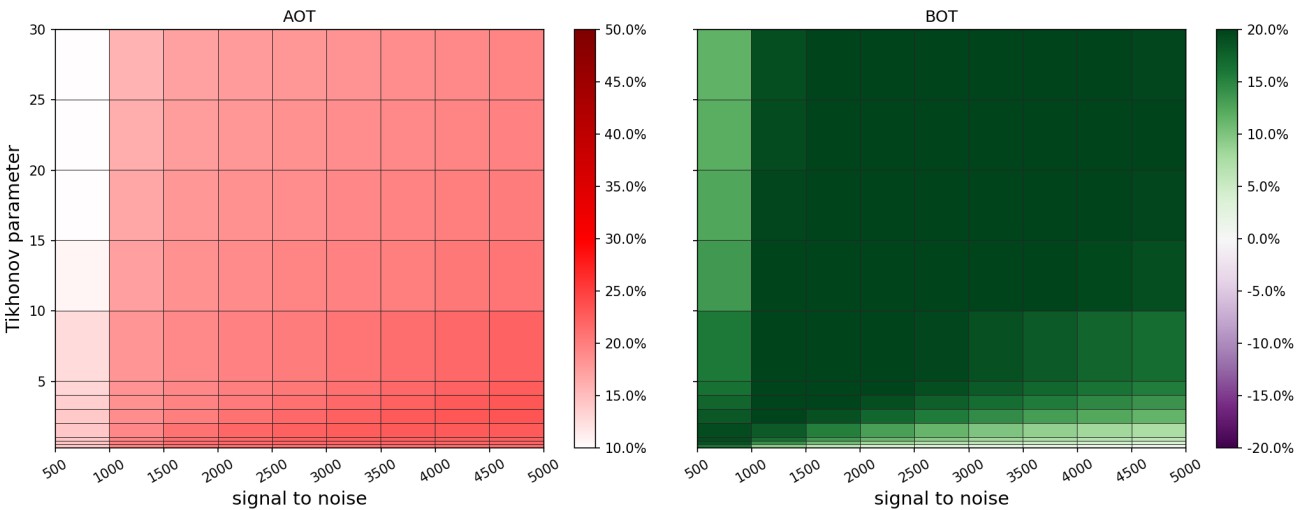

**Figure A4.** Same as figure A3 but for scenario B2.

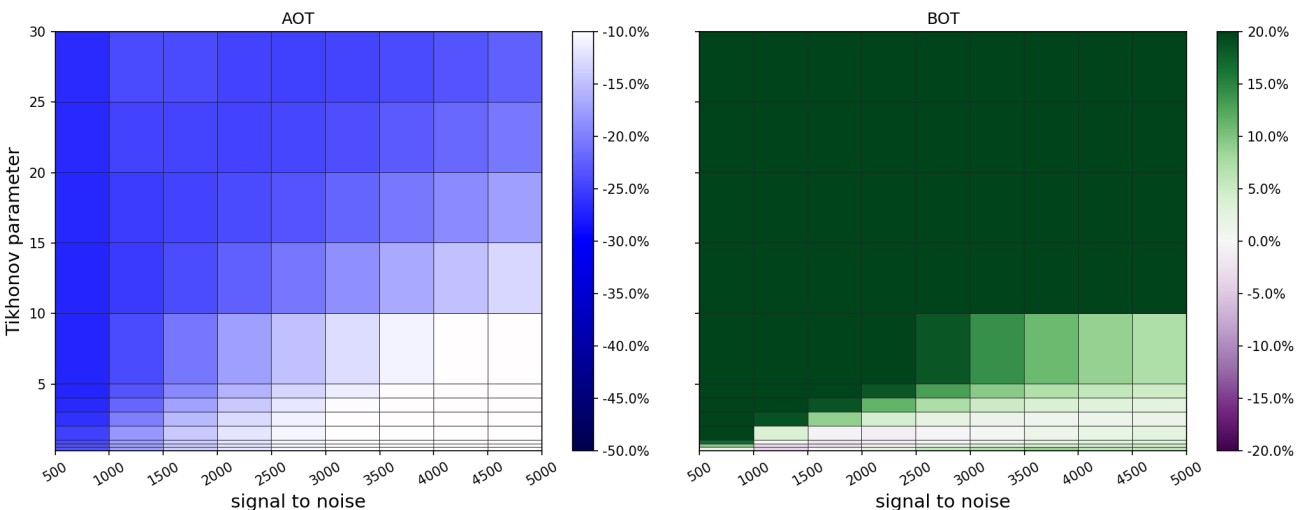

**Figure A5.** Same as Figure A3 but for scenario B3.





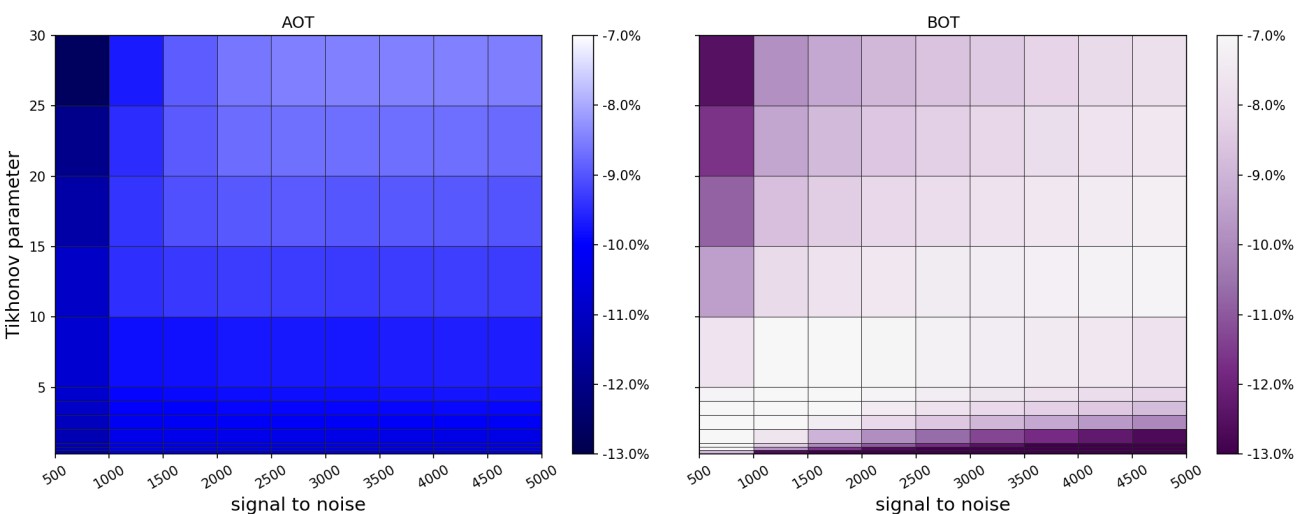

**Figure A6.** Same as Figure 5 but for scenario E1.

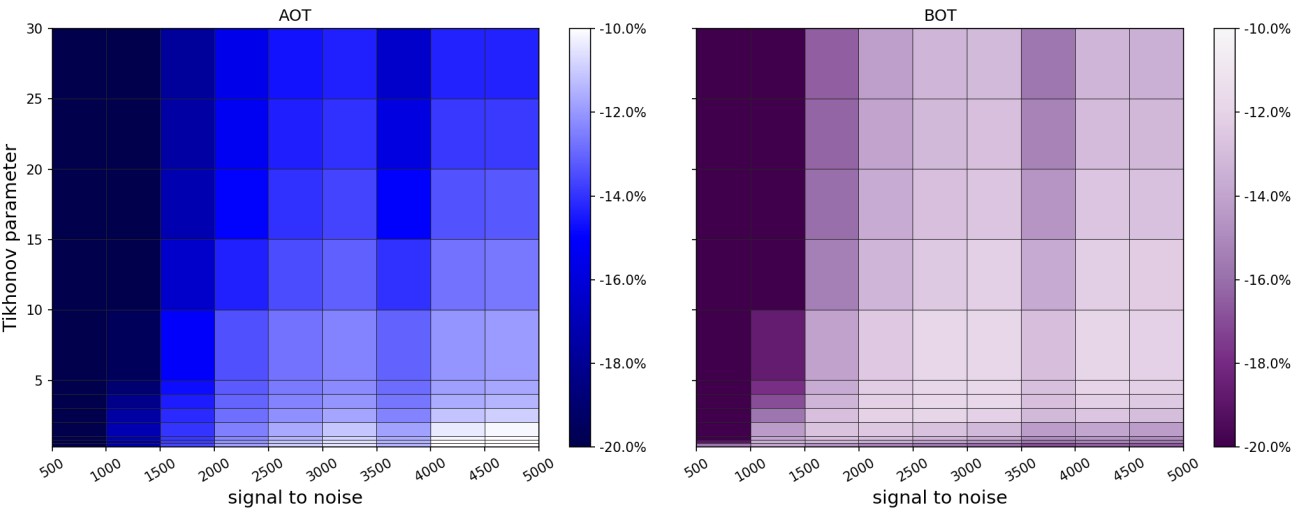

**Figure A7.** Same as Figure 5 but for scenario E2.

## A2 Additional NO$_2$ scenarios and plots

Figure A8 shows the box NO$_2$ scenarios which are retrieved for different g values in Figure A9. The standard regularisation ratio for $g = 1$ is again found to be oscillating without reaching the proper bottom concentrations for E2 and E3. Higher g factors show an improvement for both, fix a priori and pre-scaled initial guess.



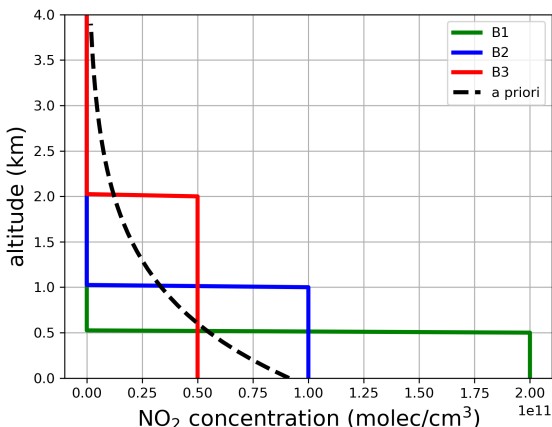

**Figure A8.** Box $NO_2$ profiles for the synthetic sensitivity study. Also shown, the a priori profile for the aerosol retrieval (dashed line).

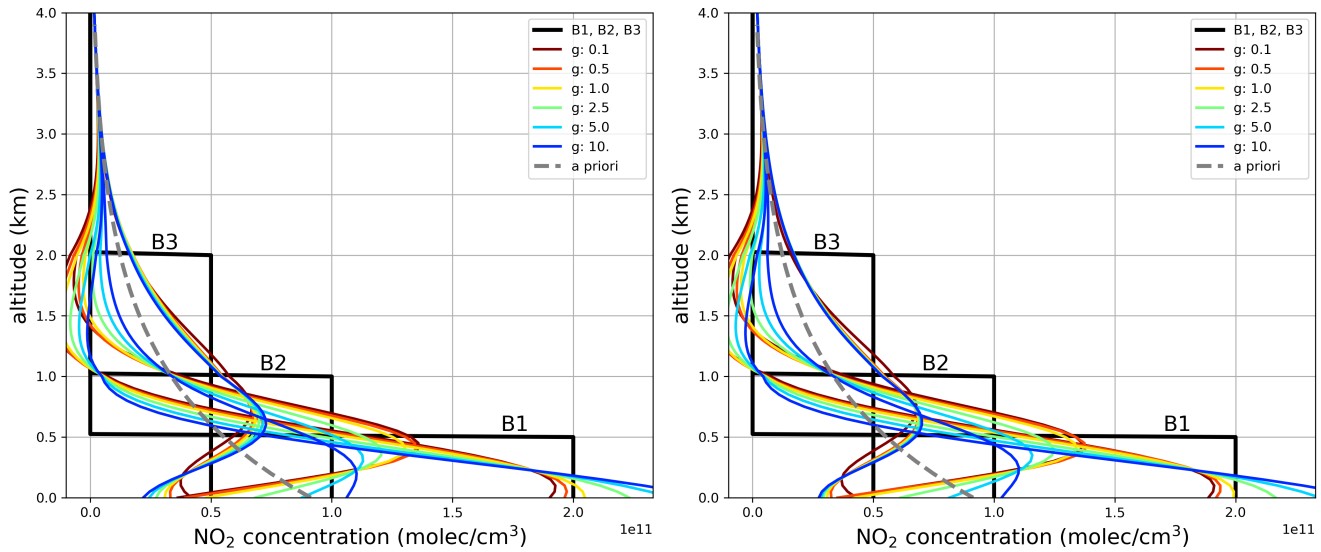

**Figure A9.** Retrieval results with a fix (left) and a pre-scaled a priori profile (right) with g factors. Small g values mean less measurement weighting for the resulting profiles.

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
