# Peer review of "BOREAS - a new MAX-DOAS profile retrieval algorithm for aerosols and trace gases"

_Atmospheric Measurement Techniques, 2018_

## Referee Comment (RC1) · Anonymous Referee #2 · 2 Aug 2018

This paper reports a new MAX-DOAS profiling algorithm detailedly. The algorithm is based on a scientific and reasonable method. The results have good correlation with the results from the other instruments. However, from the paper, it appears that the algorithm is similar to the other profiling algorithms based on the optimal estimation method.

The title of this paper is about a NEW algorithm, so you should highlight what is really NEW and innovative in your algorithm, and what are the advantages comparing to the other MAX-DOAS profiling algorithms. These points should also be included in the Abstracts.

In Figure 4, readers cannot easily distinguish between the curves corresponding to E2 and E3. Different line styles should be used for the curves of different episodes (also

in the other similar figures). In the right-hand chart, the pre-scaled a priori profiles ought to be depicted. Moreover, I cannot understand about how are the a priori profiles pre-scaled.

In the chapter about CINDI-2 campaign (4.3), the results are compared with the results from other instruments. However, it is also important to compare with the MAX-DOAS result from the same instrument but retrieved with the other algorithms.

Some well-known and fundamental introductions can be simplified. For example, it is unnecessary to explain the definition of single scattering albedo in detail and show the equation.

In the description of the algorithm, it is better to use the symbols that are commonly used in the related papers. For example, in Equation (3), it is better to use "AMF" instead of "M", "SCD" instead of "S", and "VCD" instead of "V". In Equation (5), you can just use "BAMF" instead of the curlicue M.

Minor corrections:

Page 1, Line 4: "in a second part" -> "in the second part".

Page 9, Line 5: If $\Delta y$ is a vector, then it should be described as "has N·M elements" instead of "has the dimension N·M$\times$1"; otherwise it should be described as a matrix. Similarly, Line 10, the state vector $\Delta x$ should have "L elements". Line 11, "N·M$\times$L" -> "(N·M)$\times$L".

Page 20, Line 14: "CINDI2" -> "CINDI-2".
* * *

---

## Referee Comment (RC2) · Anonymous Referee #3 · 8 Oct 2018

General comments In this paper, Tim Bösch et al. presented a new MAX-DOAS profile inversion algorithm, named as BOREAS. As the author noted, the algorithm could be the first one which retrieve profiles based on optical depths of absorbers. The algorithm is well verified through sensitivity tests with synthetic data and comparisons with various collocated independent data during the CINDI-2 campaign. In general the scientific topic is meaningful. However I think the authors should clarify two major points before publication: 1) The unique feature of the BOREAS algorithm is doing inversion based on optical depths of absorbers. The approach allows including DSOT at different wavelengths in a profile inversion. However the authors do not discuss the improvement of doing inversion with optical depths compared to with slant column densities. If there is no a considerable improvement, innovativeness of the algorithm

is doubtable. 2) In section 4, the author demonstrates that profile inversion can be improved if a priori shape is used to scale a pre-calculated AOT and VCD. However the improvement only works when a priori profile is similar with the true atmospheric profile. How do we know whether or not the a-priori shape is close to the real atmospheric profile for real measurements? If a wrong priori shape is assumed, does the method even cause larger deviations of retrieved profiles and AOT from the truth?

Specific Comments: 1) In section 3, the authors do not clarify that how the algorithm deals with negative values which could be retrieved. 2) Line 10 on page 11: aerosols can impact the sensitivities of MAX-DOAS measurements to trace gas profiles, especially at high altitudes. The optimal settings of inversion parameters could depend on aerosols. Therefore sensitivity tests should also be done under typical aerosol conditions, especially a heavy aerosol load. 3) Figure 4 (left): It is hard to identify which color curves are corresponding to individual true profiles. Please try to mark them. 4) Section 4.2: The author claim that total error of aerosol and trace gas profile inversion contains the three parts. However the level of converge, namely differences of modelled and measured SOT, could also contribute some errors. The converge level also depends on the maximum number of iterations. 5) Why is the smoothing error smaller at altitudes > 2km than that near the surface? We can expect higher uncertainties at high altitudes because MAX-DOAS profile inversion is not sensitive to high latitudes well. 6) Line 5 on page 22: The definition of a-priori variance actually increase constraint of a-priori in the inversion at high altitudes. Do you follow the same definition in the synthetic test? The definition could impact the conclusions of discussions on the optimal settings of Tikhonov, a-priori shape scaling. In addition the definition could also cause that the inversion can not retrieve lifted layers of aerosols and trace gases well. Actually the problem can be seen in Fig. 16 and Fig. 19.
* * *

---

## Author Response (AR1)

November 12, 2018

**Revised manuscript**

Dear Dr. Michel Van Roozendael,

we would like to take the opportunity to thank you for your efforts and that you accepted the editorship of our manuscript "BOREAS - a new MAX-DOAS profile retrieval algorithm for aerosols and trace gases".

Please find enclosed a revised version of our manuscript where we implemented comments by the referees. We revised the original manuscript according to their suggestions and provided also additional information as requested by the referees. We would like to mention that we also changed the data shown in the correlation plots of Figures 13, 14, 17 and 18 because we accidentally included the 12th of September. This day was initially removed from the plots. The new correlation plots correspond to the depicted time series now.

Below, the author comments are provided that have already been uploaded to the AMT web page on 05 November 2018. We also attached a new version of the revised manuscript in which changes in comparison to the initial version are highlighted. We hope that with the submission of the author comments and the revision of the manuscript, our article will be accepted for publication in AMT.

Yours sincerely,
Tim Bösch (on behalf of the co-authors)

List of Attachments

- Author comments to Referee #1
- Author comments to Referee #2
- Revised manuscript with highlighted changes
- Revised manuscript

**Author reply to Referee #2**

Tim Bösch et al.

November 5, 2018

We thank Referee #1 for carefully reading our manuscript and for the helpful comments which will improve the quality of our manuscript. We will reply to the comments point by point.

> Legend:
> - referee comments
> - author comments
> - old text from the manuscript
> - **changed text in the manuscript**

This paper reports a new MAX-DOAS profiling algorithm detailedly. The algorithm is based on a scientific reasonable method. The results have good correlation with the results from the other instruments.

Thank you very much for the positive comment.

However, from the paper, it appears that the algorithm is similar to the other profiling algorithms based on the optimal estimation method. The title of this paper is about a NEW algorithm, so you should highlight what is really NEW and innovative in your algorithm, and what are the advantages comparing to the other MAX-DOAS profiling algorithms. These points should also be included in the Abstracts.

To the authors knowledge, the approach of the aerosol retrieval is completely new. The trace gas retrieval, presented in this manuscript, is based on the optimal estimation (OE) method and is similar to retrievals used in other studies. However, the applied approach on improving the regularization ratio empirically as well as the a priori pre-scaling option are new and might help the community to reduce oscillations and wrongly weighted a priori constraints.

We will follow the suggestion of the reviewer by adding the following lines to the abstract:

**The aerosol profile retrieval is based on a novel approach in which the absorption depth of $O_4$ is directly used in order to retrieve extinction coefficient profiles instead of the commonly used perturbation theory method. The retrieval of trace gases is done with the frequently used optimal estimation method but significant improvements on how to deal with wrongly weighted a priori constraints and for scenarios in which the a priori profile is inaccurate are presented.**

Furthermore, the latter point will be also mentioned in the Introduction:

**In order to improve the standard OE method for trace gases, a simple way of changing the weighting between a priori and measurement constraints is introduced by varying a regularization factor so that oscillations are minimized. Additionally, the usage of a priori pre-scaling will be highlighted as one way of improving results when the a priori profile is far away from the real atmosphere.**

In Figure 4, readers cannot easily distinguish between the curves corresponding to E2 and E3. Different line styles should be used for the curves of different episodes (also in the other similar figures). In the right-hand chart, the pre-scaled a priori profiles ought to be depicted.

As suggested, the linestyles were changed and the pre-scaled a priori profiles were included. The following four figures were adapted for this purpose and will be added to the new manuscript.

[Figure]

Figure 1: **(Changed version of manuscript Figure 4)** Retrieval results with a fixed (left) and a pre-scaled a priori profile (right) with varying Tikhonov parameters $\gamma$ for SNR = 3000. Small $\gamma$ values mean less smoothing of the resulting profiles.

[Figure]

Figure 2: **(Changed version of manuscript Figure 7)** Retrieval results with a fix (left) and a pre-scaled a priori profile (right) with g factors. Small g values mean less measurement weighting for the resulting profiles.

[Figure]

Figure 3: **(Changed version of manuscript Figure A2)** Retrieval results with a fix (left) and a pre-scaled a priori profile (right) with varying Tikhonov parameters $\gamma$ for SNR = 3000. Small $\gamma$ values mean less smoothing of the resulting profiles.

[Figure]

Figure 4: **(Changed version of manuscript Figure A9)** Retrieval results with a fix (left) and a pre-scaled a priori profile (right) with g factors. Small g values mean less measurement weighting for the resulting profiles.

Moreover, I cannot understand about how are the a priori profiles pre-scaled.

For both, the aerosol and the trace gas retrieval, the corresponding a priori profiles are scaled so that the AOT and VCD of the new a priori profiles match the values found from an initial calculation with a "zero"-profile and from the 30° dSCD measurement for aerosols and trace gases, respectively.
In order to clarify this approach, the aerosol pre-scaling description will be extended by the following lines in Section 4.1.1:

**In Equation 15, the a priori profile $x_0$ is set to 0 but the remaining terms stay unchanged in order to retrieve a reliable AOT value without limiting the solution by a certain profile shape.**

The AOT from this pre-calculation is used to scale the a priori profile for the true run.

**The AOT from this pre-calculation is used to scale the a priori profile for the main run by calculating a scaling factor from the initial a priori AOT and the new value which is applied to all initial extinction coefficients before the main inversion.**

In Section 4.1.2, the following lines will be changed in the trace gas pre-scaling description:

This vertical column density V is used as a pre-scaling values for the a priori profile.

**The new a priori profile is scaled to match this vertical column density V by calculating a scaling factor from the initial and the retrieved V which is applied to all initial concentration values before the main inversion.**

In the chapter about CINDI-2 campaign (4.3), the results are compared with the results from other instruments. However, it is also important to compare with the MAX-DOAS result from the same instrument but retrieved with the other algorithms.

We fully agree with the reviewers suggestion. However, the BOREAS algorithm participated in a comparison study for synthetic data and real data which will pe published by Frieß et al. (2018) and Tirpitz et al. (2018), respectively. In both studies, several community algorithms are presented and compared to each other so that we decided to focus only on a validation study with ancillary measurements instead of repeating comparisons shown in the other two papers.

Some well-known and fundamental introductions can be simplified. For example, it is unnecessary to explain the definition of single scattering albedo in detail and show the equation.

We follow the reviewers suggestion by removing the definition of single scattering albedo and slant column density from the manuscript.

In the description of the algorithm, it is better to use the symbols that are commonly used in the related papers. For example, in Equation (3), it is better to use "AMF" instead of "M", "SCD" instead of "S", and "VCD" instead of "V". In Equation (5), you can just use "BAMF" instead of the curlicue M.

The authors believe that one letter variable symbols make the understanding of longer equations easier. Furthermore, these variables are also used in the literature (cf. Platt, 1994) and in other publications (cf. e.g. Marquard et al., 2000). One exception is $M$ for the air mass factor which was chosen because $A$ is additionally used for the averaging kernels.

**Minor corrections:**

Page 1, Line 4: "in a second part" $\rightarrow$ "in the second part".

Will be changed as suggested.

Page 9, Line 5: If $\Delta y$ is a vector, then it should be described as "has N·M elements" instead of "has the dimension N·M×1"; otherwise it should be described as a matrix.

Will be changed as suggested.

Similarly, Line 10, the state vector $\Delta x$ should have "L elements". Line 11, "N·M×L" $\rightarrow$ "(N·M)×L".

Will be changed as suggested.

Page 20, Line 14: "CINDI2" → "CINDI-2".

Will be changed as suggested.

November 5, 2018

We thank Referee #3 for carefully reading our manuscript and for the helpful comments which will improve the quality of our manuscript. We will reply to the comments point by point.

Legend:
- referee comments
- author comments
- old text from the manuscript
- **changed text in the manuscript**

In this paper, Tim Bösch et al. presented a new MAX-DOAS profile inversion algorithm, named as BOREAS. As the author noted, the algorithm could be the first one which retrieve profiles based on optical depths of absorbers. The algorithm is well verified through sensitivity tests with synthetic data and comparisons with various collocated independent data during the CINDI-2 campaign. In general the scientific topic is meaningful.

Thank you very much for the positive comment.

**However I think the authors should clarify two major points before publication:**

1) The unique feature of the BOREAS algorithm is doing inversion based on optical depths of absorbers. The approach allows including DSOT at different wavelengths in a profile inversion. However the authors do not discuss the improvement of doing inversion with optical depths compared to with slant column densities. If there is no a considerable improvement, innovativeness of the algorithm is doubtable.

The authors would like to clarify that inversion with optical depths is only applied to the aerosol retrieval. In order to make this clear, the following lines will be added to the abstract:
**The aerosol profile retrieval is based on a novel approach in which the absorption depth of $O_4$ is directly used in order to retrieve extinction coefficient profiles instead of the commonly used perturbation theory method. The retrieval of trace gases is done with the frequently used optimal estimation method but significant improvements on how to deal with wrongly weighted a priori constraints and for scenarios in which the a priori profile is inaccurate are presented.**
The aerosol approach was chosen in order to avoid the standard perturbation theory method. We believe that the perturbation of an a priori profile, the calculation of the weighting function matrix, the comparison of model and measured dSCD as well as the subsequent perturbation of the next layer, are time consuming and maybe complex steps which can be avoided when utilizing the optical depths. However, it should be noted that for simple atmospheric conditions, larger differences of the profiling results to other community algorithms are not expected. The generally good agreement of algorithms will be presented in the upcoming papers of Frieß et al. (2018) and Tirpitz et al. (2018).

2) In section 4, the author demonstrates that profile inversion can be improved if a priori shape is used to scale a pre-calculated AOT and VCD. However the improvement only works when a priori profile is similar with the true atmospheric profile. How do we know whether or not the a-priori shape is close to the real atmospheric profile for real measurements? If a wrong priori shape is assumed, does the method

even cause larger deviations of retrieved profiles and AOT from the truth?

The knowledge of the true atmospheric profiles is not necessary in order to apply a priori pre-scaling. If the true profile is similarly shaped as the a priori profile, a clear improvement is achieved. When the true profile has a different shape, the pre-scaling does not deteriorate the results. This can be seen in Figure A2 on the example of box profiles for aerosols and in Figure A9 for $NO_2$. Especially the B3 scenario shape is not even close to an exponential profile. Even though no clear improvement can be observed, no deterioration can be seen either. Additionally, the number of iterations for the aerosol retrieval is reduced when using an a priori profile closer to the true atmospheric conditions.
In order to clarify this, the following lines will be adapted in Section 4.1.1 in the manuscript:

However, no improvement is found when the shape of the a priori profile deviates strongly from the true atmospheric condition (see box profiles in appendix A).
**However, no improvement is found when the shape of the a priori profile deviates strongly from the true atmospheric condition but no deterioration either (see box profiles in appendix A).**

**Specific Comments:**

1) In section 3, the authors do not clarify that how the algorithm deals with negative values which could be retrieved.

The aerosol profile retrieval works in the logarithmic space and thus no negative values can be retrieved. The optimal estimation based trace gas retrieval works in the linear space. Here, negative values are possible, especially when the regularization ratio is wrongly weighted. However, a priori pre scaling leads to a better weighting of a priori constrains in addition to the usage of a regularization parameter $g \neq 1$. Negative values are still possible but rare. While negative values are meaningless from a chemical/physical perspective they can be understood as a representation of uncertainties in the respective altitude region from a mathematical point of view.
The following line will be added to Section 3.1 in the manuscript:

**Equation 15 is solved in the logarithmic space in order to avoid negative values.**

Additionally, Section 3.2 will be adapted:

**Equation 18 is solved in the linear space. Negative values are possible but can be avoided by applying appropriate a priori constraints.**

2) Line 10 on page 11: aerosols can impact the sensitivities of MAX-DOAS measurements to trace gas profiles, especially at high altitudes. The optimal settings of inversion parameters could depend on aerosols. Therefore sensitivity tests should also be done under typical aerosol conditions, especially a heavy aerosol load.

We fully agree with the reviewers suggestion. However, the BOREAS algorithm participated in a comparison study for synthetic data and real data which will pe published by Frieß et al. (2018) and Tirpitz et al. (2018), respectively. Especially in the first study, various combinations of aerosol and trace gas profiles (also with a heavy aerosol load) are investigated in order to answer this question. Therefore, we would like to avoid repeating similar investigations on this matter in this manuscript.

3) Figure 4 (left): It is hard to identify which color curves are corresponding to individual true profiles. Please try to mark them.

As suggested, the linestyles were changed and the pre-scaled a priori profiles were included (as asked by

reviewer #2). The following four figures were adapted for this purpose and will be added to the revised manuscript.

[Figure]

Figure 1: **(Changed version of manuscript Figure 4)** Retrieval results with a fixed (left) and a pre-scaled a priori profile (right) with varying Tikhonov parameters $\gamma$ for SNR = 3000. Small $\gamma$ values mean less smoothing of the resulting profiles.

[Figure]

Figure 2: **(Changed version of manuscript Figure 7)** Retrieval results with a fix (left) and a pre-scaled a priori profile (right) with g factors. Small g values mean less measurement weighting for the resulting profiles.

[Figure]

Figure 3: **(Changed version of manuscript Figure A2)** Retrieval results with a fix (left) and a pre-scaled a priori profile (right) with varying Tikhonov parameters $\gamma$ for SNR = 3000. Small $\gamma$ values mean less smoothing of the resulting profiles.

[Figure]

Figure 4: **(Changed version of manuscript Figure A9)** Retrieval results with a fix (left) and a pre-scaled a priori profile (right) with g factors. Small g values mean less measurement weighting for the resulting profiles.

4) Section 4.2: The author claim that total error of aerosol and trace gas profile inversion contains the three parts. However the level of converge, namely differences of modelled and measured SOT, could also contribute some errors. The converge level also depends on the maximum number of iterations.

The RMS between measured and simulated dSOT or dSCD differences for the aerosol or trace gas retrieval, respectively, is a complicated quantity as it does not say a lot about the quality of the retrieved profiles. If constraints on the measurement are chosen too weak, very good agreement can be achieved between measured and modelled slant columns but the resulting profiles often show strong oscillation because of the ill-posed problem of profile inversion. Local minima in the respective RMS quantities exist but their analysis is, especially for aerosols, time consuming and the interpretation depends strongly on the actual atmospheric conditions. Due to this issue, we do not suggest the usage of these quantities for an error analysis with respect to the quality of the retrieved profiles.

5) Why is the smoothing error smaller at altitudes > 2 km than that near the surface? We can expect higher uncertainties at high altitudes because MAX-DOAS profile inversion is not sensitive to high latitudes well.

For the retrieval of aerosol profiles, higher altitude a priori variances were set to smaller values in order to reduce issues near the grid boundary where the sensitivity is lowest but small instabilities might lead to strong oscillations. The small errors should not be misinterpreted as lower uncertainties in the specific altitudes but are deliberately suppressed.

The following lines will be added to Section 4.2 in the manuscript to explain this issue:

**Note that the errors are nearly zero for higher altitudes because of the height depending variances which were chosen in order to reduce possible retrieval instabilities in altitude regimes where the sensitivity is lowest.**

6) Line 5 on page 22: The definition of a-priori variance actually increase constraint of a-priori in the inversion at high altitudes. Do you follow the same definition in the synthetic test? The definition could impact the conclusions of discussions on the optimal settings of Tikhonov, a-priori shape scaling.

For the depicted synthetic scenarios a similar definition was chosen. We agree that this is an additional constraint but the optimal Tikhonov parameter is generally not strongly affected by this choice as the variances are only of minor importance when changing the Tikhonov parameter drastically. The a priori pre-scaling is not negatively affected by this choice. Also a constant variance would yield in a clear improvement under the assumptions discussed in the corresponding sections (4.1.1, 4.1.2).

In addition the definition could also cause that the inversion can not retrieve lifted layers of aerosols and trace gases well. Actually the problem can be seen in Fig. 16 and Fig. 19.

We agree partially. The definition reduces the sensitivity in higher altitudes as we force the result stronger into the direction of the a priori profile. With the assumption of an exponentially shaped a priori, we further assumed the main load/concentration to be close to the surface. While the aerosol retrieval is still able to retrieve elevated layers due to the applied iterations, the trace gas retrieval will fail in retrieving the proper maximum value and altitude. However, in theses cases, the profiling results will still show elevated features as can be seen in Figure 16 but not in Figure 19. Unfortunately, both Figures show two difficulties. Figure 16 depicts elevated layers (08:00 and 10:00 UTC) in altitudes where the chance of a successful retrieval is low, which is independent from the a priori variance. Only the change to another a priori profile would result in a clearly better retrieval response. On the other hand, Figure 19 shows elevated layers with such a small vertical extent that the vertical resolution of MAX-DOAS profiling is simply not sufficient.

The following lines will be added to the manuscript (Section 4.3) in order to clarify that the choice of a height depending variance is of minor importance for the results:

[revised manuscript text omitted]